# *Connect, Collapse, Corrupt:* LEARNING CROSS-MODAL TASKS WITH UNI-MODAL DATA

**Yuhui Zhang**[*], **Elaine Sui**[*], **Serena Yeung-Levy**
Stanford University
{yuhuiz, esui, syyeung}@cs.stanford.edu

## ABSTRACT

Building cross-modal applications is challenging due to limited paired multi-modal data. Recent works have shown that leveraging a pre-trained multi-modal contrastive representation space enables cross-modal tasks to be learned from uni-modal data. This is based on the assumption that contrastive optimization makes embeddings from different modalities interchangeable. However, this assumption is under-explored due to the poorly understood geometry of the multi-modal contrastive space, where a modality gap exists. In our study, we provide a theoretical explanation of this space's geometry and introduce a three-step method, $C^3$ (Connect, Collapse, Corrupt), to bridge the modality gap, enhancing the interchangeability of embeddings. Our $C^3$ method significantly improves cross-modal learning from uni-modal data, achieving state-of-the-art results on zero-shot image / audio / video captioning and text-to-image generation.

## 1 INTRODUCTION

Building cross-modal applications often face a critical challenge: the scarcity of paired multi-modal data. Large-scale multi-modal contrastive learning has emerged as a promising approach to address this gap. Pioneering works like CLIP (Radford et al., 2021) and ImageBind (Girdhar et al., 2023) offer publicly available representation spaces learned from contrasting millions of web-based images and texts. These spaces that align embeddings of concepts across different modalities pave the way for leveraging abundant uni-modal data in cross-modal tasks. For instance, instead of training an image captioning system on image embeddings, one could train it on caption embeddings. Likewise, text-to-image generation models can utilize image embeddings as opposed to text. During cross-modal inference, embeddings from the other modality are simply input into the model (Figure 1). Notably, this approach has achieved impressive results in image captioning (Li et al., 2023; Tam et al., 2023; Nukrai et al., 2022) and text-to-image generation (Zhou et al., 2022c;a;b), eliminating the need for multi-modal paired data.

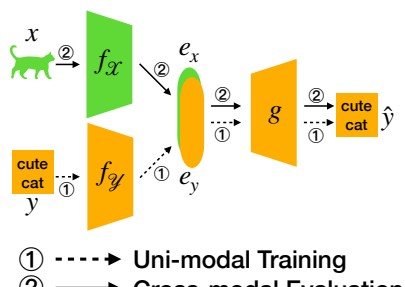

Figure 1: Interchangeable use of embeddings enables learning cross-modal tasks with uni-modal data.

This aforementioned process is based on the hypothesis that image embeddings and text embeddings can be used interchangeably in the multi-modal representation space. However, the validity of this assumption is not well understood. Recent works reveal that the resulting geometry from multi-modal contrastive learning is nontrivial — corresponding image and text embeddings do not necessarily collapse to the same points in the space. Instead, there is a significant gap between embeddings from different modalities, potentially hindering the direct interchangeable use of image and text embeddings (Liang et al., 2022; Zhang et al., 2023). The lack of comprehension of the resulting geometry from contrastive learning makes it challenging to design methods to leverage this representation space and learn cross-modal tasks from uni-modal data.

---

[*]Equal contribution. Project page available at https://yuhui-zh15.github.io/C3-Website/.

In this study, we conclude that the geometry of multi-modal contrastive representation space as (Figure 2):

$$\boldsymbol{e}_x - \boldsymbol{e}_y = \boldsymbol{c}_\perp + \boldsymbol{\epsilon}$$

where $\boldsymbol{e}_x$ and $\boldsymbol{e}_y$ denote the embeddings of paired inputs from different modalities, $\boldsymbol{c}_\perp$ is a constant vector representing the modality gap, which is orthogonal to the embedding span of $\boldsymbol{e}_x$ and $\boldsymbol{e}_y$, and $\epsilon$ is a random vector representing the alignment noise, which can be approximated by a Gaussian distribution.

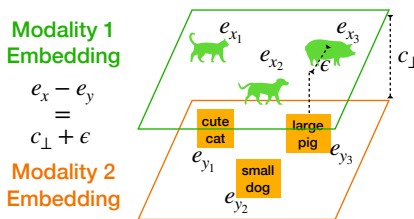

Figure 2: Geometry of the multi-modal contrastive representation space.

We provide a theoretical explanation of the above geometry. Specifically, the modality gap emerges during initialization as certain dimensions of image and text embeddings remain approximately constant in the embedding space, and the constants are distinct for images and texts due to separate initializations. During optimization, these constant dimensions lack a gradient that pushes them to align to the same value, leading to the preservation and orthogonality of the modality gap. Meanwhile, the alignment noise arises from the stable region produced by the contrastive loss, where points within a certain range result in a loss near zero, causing the optimization to halt.

Based on this understanding of the geometry, we propose a simple method, $C^3$, to improve the interchangeability of embeddings from different modalities, thereby enabling the learning of cross-modal tasks with uni-modal data. $C^3$ consists of three steps:

1. Connect: Related concepts from different modalities are connected via multi-modal contrastive learning, resulting in a shared representation space that can allow the interchangeability of embeddings from the different modalities. However, a modality gap and alignment noise exist.
2. Collapse: Based on its geometry, the modality gap can be closed by subtracting the embedding mean from each modality, eliminating the most dominant distributional difference between them.
3. Corrupt: Additional noise is introduced as regularization during training to improve the performance and robustness of learning cross-modal tasks from uni-modal data, given the alignment noise from multi-modal contrastive learning.

We demonstrate the effectiveness and broad generalization of $C^3$ on four tasks: image, audio, video captioning and text-to-image generation, and achieve state-of-the-art performance on zero-shot evaluation settings when trained solely on uni-modal data. We also provide a detailed analysis of each component that contributes to performance improvements. Our method sheds new light on the possibility of achieving cross-modal tasks under a low-data regime, and our theoretical analysis provides insight into understanding multi-modal contrastive learning.

Our contributions are three-fold:

1. We provide a theoretical explanation of the representation space geometry resulting from multi-modal contrastive learning.
2. Based on this geometry, we propose a simple three-step solution to enhance the interchangeability of embeddings from different modalities, improving cross-modal learning with uni-modal data.
3. We show the effectiveness of our method on image / audio / video captioning and text-to-image generation, achieving state-of-the-art results.

## 2 LEARNING CROSS-MODAL TASKS WITH UNI-MODAL DATA

In this section, we present the general notion of a cross-modal task and how we can leverage uni-modal data to learn such tasks.

### 2.1 CROSS-MODAL TASK FORMULATION

Cross-modal tasks aim to learn a model that maps inputs from one modality $\mathcal{X}$ (e.g., images) to another modality $\mathcal{Y}$ (e.g., texts). Given a paired multi-modal dataset $\mathcal{D} = \{(x, y) \in \mathcal{X} \times \mathcal{Y}\}$ (e.g., an image-caption dataset), the task can be achieved by minimizing the empirical risk $\mathcal{L}_d$ between the predicted target $\hat{y} = g(f_\mathcal{X}(x))$ and the true target $y$ over the dataset $\mathcal{D}$, where $f_\mathcal{X} : \mathcal{X} \rightarrow \mathbb{R}^d$ is an encoder that maps inputs from $\mathcal{X}$ to a $d$-dimensional representation space, and $g : \mathbb{R}^d \rightarrow \mathcal{Y}$ is a

decoder that maps outputs from the encoder to $\mathcal{Y}$:

$$\min_{g, f_\mathcal{X}} \frac{1}{|\mathcal{D}|} \sum_{(x,y) \in \mathcal{D}} \mathcal{L}_d(\hat{y}, y), \quad \hat{y} = g(f_\mathcal{X}(x))$$

$\mathcal{L}_d$ measures the discrepancy between the predicted and true targets, which can be mean squared error (MSE) for images and cross-entropy loss for texts.

## 2.2 ENABLING CROSS-MODAL TASKS WITH UNI-MODAL DATA

The need for a multi-modal paired dataset $\mathcal{D}$ to learn cross-modal tasks is suboptimal, as collecting such datasets can be expensive and time-consuming. However, if we have a encoder $f_\mathcal{Y} : \mathcal{Y} \to \mathbb{R}^d$ that maps inputs from $\mathcal{Y}$ to the same representation space as the encoder $f_\mathcal{X}$, i.e., $\forall x, y \in \mathcal{D}, f_\mathcal{X}(x) = f_\mathcal{Y}(y)$, we can train the cross-modal task using a uni-modal dataset $\mathcal{D}' = \{y \in \mathcal{Y}\}$:

$$\min_g \frac{1}{|\mathcal{D}'|} \sum_{y \in \mathcal{D}'} \mathcal{L}_d(\hat{y}, y), \quad \hat{y} = g(f_\mathcal{Y}(y))$$

Note that $f_\mathcal{Y}$ should be frozen during training to maintain its embedding alignment with $f_\mathcal{X}$. When evaluating in a cross-modal setting, we can replace $f_\mathcal{X}$ with $f_\mathcal{Y}$, thus enabling cross-modal tasks with only uni-modal training data.

## 2.3 ESTABLISHING A SHARED REPRESENTATION SPACE

Recent advances in multi-modal contrastive learning has enabled for encoders that map similar inputs from different modalities to a shared representation space. Specifically, given a large multi-modal dataset[1], $n$ paired inputs are randomly sampled during each iteration and the following objective is optimized (Radford et al., 2021):

$$\min_{f_\mathcal{X}, f_\mathcal{Y}} \mathcal{L} = -\frac{1}{2n} \sum_{i=1}^n \left( \log \frac{\exp(\boldsymbol{e}_{x_i} \cdot \boldsymbol{e}_{y_i}/\tau)}{\sum_{j=1}^n \exp(\boldsymbol{e}_{x_i} \cdot \boldsymbol{e}_{y_j}/\tau)} + \log \frac{\exp(\boldsymbol{e}_{x_i} \cdot \boldsymbol{e}_{y_i}/\tau)}{\sum_{j=1}^n \exp(\boldsymbol{e}_{x_j} \cdot \boldsymbol{e}_{y_i}/\tau)} \right)$$

where $\boldsymbol{e}_x = f_\mathcal{X}(x)$, $\boldsymbol{e}_y = f_\mathcal{Y}(y)$, and $\tau$ is the temperature hyperparameter. This loss function encourages high similarity between the embeddings of the paired image-texts relative to the similarities between unpaired ones. Intuitively, after optimizing the loss, paired image and text embeddings should collapse to the same point. However, empirical results have shown a significant gap between paired embeddings (Liang et al., 2022), which prevents the direct interchangeable use of image and text embeddings. In the next section, we provide a detailed analysis of the geometry of the multi-modal contrastive representation space.

## 3 MULTI-MODAL CONTRASTIVE REPRESENTATION SPACE GEOMETRY

We begin by providing the following proposition that describes the geometry of the multi-modal contrastive representation space. These findings inform our general method of how to adapt uni-modal data for cross-modal learning.

**Proposition 1. (Multi-modal Contrastive Representation Space Geometry)**
*Given a paired image $x$ and text $y$, the relationship between the $\ell_2$-normalized image embedding $e_x$ and text embedding $e_y$ obtained from multi-modal contrastive learning can be described as:*

$$\boldsymbol{e}_x - \boldsymbol{e}_y = \boldsymbol{c}_\perp + \boldsymbol{\epsilon}$$

*where $\boldsymbol{c}_\perp$ is a constant vector representing the modality gap and is orthogonal to the image and text embedding span, i.e., $\forall \boldsymbol{e}_{x_1}, \boldsymbol{e}_{x_2}, \boldsymbol{c}_\perp \cdot (\boldsymbol{e}_{x_1} - \boldsymbol{e}_{x_2}) = 0$; $\boldsymbol{\epsilon} \sim \mathcal{N}(\boldsymbol{0}, \sigma^2 \boldsymbol{I})$ is a random Gaussian vector representing the alignment noise.*

In the following subsections, we will first explain why the modality gap $\boldsymbol{c}_\perp$ exists before and after model optimization. Then, we will introduce the alignment noise $\boldsymbol{\epsilon}$ and its relation to the temperature parameter in contrastive loss.

---

[1] Acquiring domain-specific paired multi-modal datasets can be challenging. However, several works have gathered large-scale noisy image-caption pairs from the web and made pre-trained encoders $f_\mathcal{X}$ and $f_\mathcal{Y}$ available for direct use.

## 3.1 MODALITY GAP

The presence of a modality gap and its orthogonality to the image and text embedding span are due to the joint effect of initialization and optimization.

**Initialization.** The modality gap exists when randomly initializing multi-modal models, which can be explained by the dimensional collapse (Jing et al., 2022) of the representation space defined below.

**Definition 1. (Dimensional Collapse of the Representation Space)**
*Given a $d$-dimensional representation space $\mathbb{R}^d$, we define its underline{effective dimension} $d_e$ as:*

$$d_e = \arg\min_{d'} \frac{\sum_{i=1}^{d'} \sigma_i}{\sum_{i=1}^{d} \sigma_i} \geq \gamma$$

*where the $\sigma_i$'s are the singular values of the representation covariance matrix in decreasing order, and $\gamma$ thresholds the minimum variance explained by the $d_e$ dimensions. Dimensional collapse occurs when $d_e \ll d$.*

To demonstrate the dimensional collapse phenomenon, we took a randomly initialized CLIP with a $d = 512$ representation space and fed MS-COCO images and captions as input. We obtained the corresponding image features and text features and performed SVD on the image feature and text feature covariance matrices. The distribution of the singular values is shown in Figure 3, where we can clearly see that the effective dimension of both the image and text features is small. Specifically, when setting $\gamma = 0.99$, the effective dimension of image embeddings is $d_{e,x} = 25$ and that of text embeddings is $d_{e,y} = 230$. Therefore, the effective dimension of the shared representation

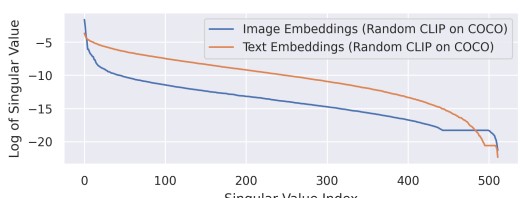

Figure 3: Dimensional collapse of the CLIP representation space. Singular values obtained from SVD reveal that the effective dimension of the image and text representation space is much smaller than the total number of dimensions.

space $d_e \leq d_{e,x} + d_{e,y} = 255$. The equality holds only when all the effective dimensions of the image and text are orthogonal.

Dimensional collapse indicates that only a small number of dimensions contribute significantly to the variance of the representation space, while the remaining dimensions can be viewed as constant. Suppose the shared representation space has a maximum effective dimension $d_e = 255$. This indicates that the image and text embeddings will remain constant in the $d_c = d - d_e = 257$ ineffective dimensions. As a result, a modality gap exists at the beginning of model optimization, as these $d_c$ ineffective dimensions will be inherently different for images and texts, given random initialization.

To verify this, we synthesize $n = 1,000$ image and text embeddings in $d = 512$ space. For each image embedding, we initialize the first $d_{e,x}$ dimensions, and for each text embedding, the $d_{e,x}$-th to $(d_{e,x}+d_{e,y})$-th dimensions, by randomly sampling a standard Gaussian distribution $\mathcal{N}(0,1)$. All the other dimensions are set to a constant value drawn from a $\mathcal{N}(0,1)$, where the constant for the image and text embeddings are different. Figure 4 (left) illustrates this setup by showing the variance of each dimension. This setup mimics our findings of SVD analysis on CLIP, where we set image embeddings to have $d_{e,x}$ effective dimensions and text embeddings to have $d_{e,y}$ effective dimensions, and assume these effective dimensions are fully orthogonal. We normalize the embeddings to unit length before performing our analysis.

We observe a clear modality gap at the beginning: the $\ell_2$-distance between the mean of the image embeddings and the mean of the text embeddings is $1.21$. When we only consider the last $d_c$ ineffective dimensions, the average distance is $0.99$. More discussion is in Appendix I.

**Optimization.** The analysis above reveals that the modality gap exists at initialization, and we further analyze why optimizing for the multi-modal contrastive loss fails to close the gap. The following lemma reveals that there is no gradient in the modality gap direction, therefore the gap and its orthogonality will be preserved.

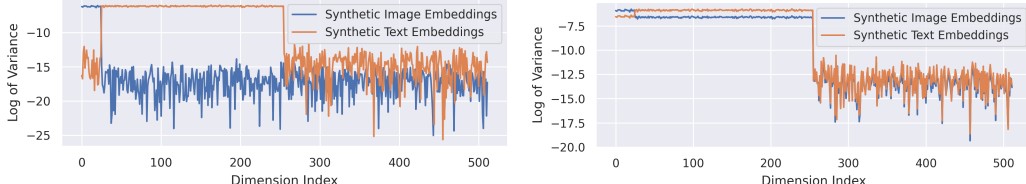

Figure 4: Variance of each dimension before (left) and after (right) multi-modal contrastive optimization. Our analysis reveals that gradients will only be propagated to effective dimensions and no gradient will be propagated to ineffective dimensions. Therefore, the effective dimensions are aligned while ineffective dimensions remain constant after optimization.

**Lemma 1. (Gradients in Contrastive Optimization)** *(Proof in Appendix B)*
*With the mild assumption of equal presence of $n$ images and texts with $p(x_i) = p(y_i) = 1/n$, optimizing the multi-modal contrastive loss $\mathcal{L} = -\frac{1}{2n}\sum_{i=1}^{n}\big(\log\frac{\exp(\boldsymbol{e}_{x_i}\cdot\boldsymbol{e}_{y_i}/\tau)}{\sum_{j=1}^{n}\exp(\boldsymbol{e}_{x_i}\cdot\boldsymbol{e}_{y_j}/\tau)} + \log\frac{\exp(\boldsymbol{e}_{x_i}\cdot\boldsymbol{e}_{y_i}/\tau)}{\sum_{j=1}^{n}\exp(\boldsymbol{e}_{x_j}\cdot\boldsymbol{e}_{y_i}/\tau)}\big)$ yields the following gradients:*

$$\nabla_{\boldsymbol{e}_{x_i}}\mathcal{L} = \lambda\sum_{j=1}^{n}\alpha_{y_j}(\boldsymbol{e}_{y_j} - \boldsymbol{e}_{y_i}), \qquad \nabla_{\boldsymbol{e}_{y_i}}\mathcal{L} = \lambda\sum_{j=1}^{n}\alpha_{x_j}(\boldsymbol{e}_{x_j} - \boldsymbol{e}_{x_i})$$

*where $\lambda = 1/(2n\tau)$, $\alpha_{x_j} = p(x_j|y_i) + p(y_i|x_j)$, $\alpha_{y_j} = p(y_j|x_i) + p(x_i|y_j)$, $p(x_i|y_j) = \frac{\exp(\boldsymbol{e}_{x_i}\cdot\boldsymbol{e}_{y_j}/\tau)}{\sum_{k=1}^{n}\exp(\boldsymbol{e}_{x_k}\cdot\boldsymbol{e}_{y_j}/\tau)}, p(y_i|x_j) = \frac{\exp(\boldsymbol{e}_{y_i}\cdot\boldsymbol{e}_{x_j}/\tau)}{\sum_{k=1}^{n}\exp(\boldsymbol{e}_{y_k}\cdot\boldsymbol{e}_{x_j}/\tau)}$, and $\tau$ is temperature.*

Lemma 1 highlights that the gradients of image embeddings during contrastive optimization are fully determined by the text embedding span, and vice versa. Due to the dimensional collapse of the image and text embedding span, the contrastive optimization process fails to propagate gradients in the direction of the ineffective dimensions, resulting in gap preservation and orthogonality to the image and text embedding span after optimization. Lemma 1 also implies that the effective dimensionality of the joint representation space remains unchanged after optimization.

To empirically verify this, we optimize the multi-modal contrastive loss $\mathcal{L}$ on the $n = 1,000$ previously synthesized image and text embeddings. We optimize for 200K steps with a learning rate 0.1, and CLIP's initial temperature of $\tau = 0.07$. We compute the variance of each dimension pre- and post-contrastive optimization and plot the results in Figure 4 (right). From the figure, we can see that the first $d_e = 255$ effective dimensions are aligned after optimization while the last $d_c = 257$ ineffective ones remain unchanged. This verifies that no gradient is propagated to the ineffective dimensions, as the variance for these dimensions remains zero after optimization. The modality gap becomes slightly smaller (0.82 compared to 0.99 before optimization) mainly due to $\ell_2$-regularization, where changes in the effective dimensions affect changes in the ineffective ones.

In summary, due to dimensional collapse at model initialization, there are ineffective dimensions where image and text embeddings can be viewed as different constants, resulting in a modality gap at initialization. During optimization, these ineffective dimensions have no gradient update, and thus, the modality gap and its orthogonality are preserved after optimization.

### 3.2 ALIGNMENT NOISE

In this section, we explain alignment noise after multi-modal contrastive learning. This noise results from the stable region of contrastive loss demonstrated in the following lemma, where we can consider a single term in the loss given the symmetry of the added terms.

**Lemma 2. (Stable Region Controlled by Temperature)** *(Proof in Appendix B)*
*We consider a single term in the multi-modal contrastive loss $\mathcal{L}_i = -\log\frac{\exp(\boldsymbol{e}_{x_i}\cdot\boldsymbol{e}_{y_i}/\tau)}{\sum_{j=1}^{n}\exp(\boldsymbol{e}_{x_i}\cdot\boldsymbol{e}_{y_j}/\tau)}$. We define the margin $r = \boldsymbol{e}_{x_i}\cdot\boldsymbol{e}_{y_i} - \max_{j\neq i}\boldsymbol{e}_{x_i}\cdot\boldsymbol{e}_{y_j}$ as the measure of the similarity difference between the matched pair and the hardest negative pair. When $r$ exceeds a threshold given below, $\mathcal{L}_i$ falls below a small pre-set value $\delta$, where we assume optimization ends:*

$$r \geq \tau\log\frac{o(\tau)}{\exp(\delta) - 1},$$

*where $o(\tau)$ is a monotonically increasing function of temperature $\tau$ that satisfies $1 < o(\tau) < n$. Therefore, the required margin $r$ is monotonically increasing with $\tau$.*

Lemma 2 suggests that there is a stable region of the contrastive loss. This region can be viewed as a function of temperature where region size increases as the temperature decreases, with the required margin becoming smaller. Within this stable region, the loss falls below the small value $\delta$, indicating that optimization has ended. In the extreme case where $\tau \to 0_+$, if $r \geq 0$, the loss will be less than $\delta$ ($\delta \to 0_+$ for this case). This means that given all the $y_j$, $x_i$ can end up within a region instead of a fixed point, resulting in the same zero loss. Figure 5 illustrates the stable region defined by

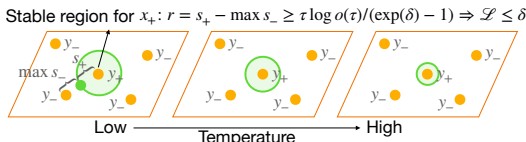

Figure 5: Stable region (green area) of contrastive learning controlled by temperature. Within the stable region, the loss falls below a small preset value, indicating that optimization has ended. The region increases as the temperature decreases.

the margin with regard to the temperature. Therefore, there may be a mismatch between the matched pairs in the representation space, resulting in alignment noise.

### 3.3 Summary

In Section 3.1 and 3.2, we explain the modality gap $\boldsymbol{c}_\perp$ and alignment noise $\boldsymbol{\epsilon}$ in Proposition 1, respectively. Combining them together, we explain the geometric relation of paired embeddings $\boldsymbol{e}_x - \boldsymbol{e}_y = \boldsymbol{c}_\perp + \boldsymbol{\epsilon}$.

| Statistic | Mean | Std |
|---|---|---|
| $\|\boldsymbol{d}^{(i)}\|_2$ | 0.83 | 0.01 |
| $\cos(\boldsymbol{d}^{(i)}, \boldsymbol{d}^{(j)})$ | 0.99 | 0.00 |
| $\cos(\boldsymbol{d}^{(i)}, \boldsymbol{r}_{j,k}^{(i)})]$ | 0.00 | 0.06 |
| $\mathbb{E}[\boldsymbol{\epsilon}_j^{(i)}]_k$ | 0.00 | 0.00 |
| $\cos(\boldsymbol{\epsilon}_j^{(i)}, \boldsymbol{\epsilon}_k^{(i)})$ | 0.00 | 0.10 |

Table 1: Statistics that reveals representation space geometry.

We verify this geometric relation using CLIP on the MS-COCO image-caption dataset. We randomly group each 100 images into group $i$, and define individual gap $\boldsymbol{d}_j^{(i)} = \boldsymbol{e}_{x_j}^{(i)} - \boldsymbol{e}_{y_j}^{(i)}$, group gap $\boldsymbol{d}^{(i)} = \mathbb{E}_j[\boldsymbol{d}_j^{(i)}]$, image difference $\boldsymbol{r}_{j,k}^{(i)} = \boldsymbol{e}_{x_j}^{(i)} - \boldsymbol{e}_{x_k}^{(i)}$, alignment noise $\boldsymbol{\epsilon}_j^{(i)} = \boldsymbol{d}_j^{(i)} - \boldsymbol{d}^{(i)}$, where $j, k$ are image or text indices. These statistics are computed in Table 1, where the first three statistics show that the modality gap $\boldsymbol{c}_\perp$ approximates a constant vector orthogonal to the image and text embedding span, and the last two statistics show that the alignment noise $\boldsymbol{\epsilon}$ can be viewed as Gaussian noise. We provide a detailed explanation of how to interpret these statistics in Appendix C.

This geometric analysis in this section serves as the foundation of the approach we introduce in the next section, where we develop a simple method to align the shared representation space and enable learning cross-modal tasks with uni-modal data.

## 4 Connect, Collapse, Corrupt

Proposition 1 from Section 3 reveals the geometry of the multi-modal contrastive representation space. Based on this, we propose three steps, connect, collapse, corrupt ($C^3$), to align the representation space, making it possible for embeddings from different modalities to be interchangeably consumed by the decoder and thus enabling learning cross-modal tasks from uni-modal data.

**Stage 1: Connect.** This stage establishes connections between similar concepts across different modalities. We leverage recent advances in multi-modal contrastive learning (Radford et al., 2021) and use encoders trained with this strategy to build cross-modal models. However, a modality gap and alignment noise exists after multi-modal contrastive learning, as shown in Proposition 1.

**Stage 2: Collapse.** When directly using embeddings of different modalities as input, there is a drastic degradation in performance due to the modality gap, which causes input distributions to the decoder to differ. To address this issue, we adopt a simple approach proposed by (Zhang et al., 2023) that effectively removes the modality gap. Specifically, during training, in place of $\boldsymbol{e}_x$, we feed in $\boldsymbol{e}_x' = \boldsymbol{e}_x - \mathbb{E}_x[\boldsymbol{e}_x]$ to the decoder, and during inference with another modality, in place of $\boldsymbol{e}_y$, we feed in $\boldsymbol{e}_y' = \boldsymbol{e}_y - \mathbb{E}_y[\boldsymbol{e}_y]$. This approach collapses the modality gap, eliminating the input distribution mismatch between the two modalities:

$$\boldsymbol{e}_x' - \boldsymbol{e}_y' = (\boldsymbol{e}_x - \boldsymbol{e}_y) - (\mathbb{E}_x[\boldsymbol{e}_x] - \mathbb{E}_y[\boldsymbol{e}_y]) = \boldsymbol{\epsilon}$$

**Stage 3: Corrupt.**  After removing the modality gap, there is still alignment noise which can be approximated by a $\mathcal{N}(0, \sigma^2 I)$. During unsupervised training, instead of directly decoding $y$ from $e'_y$, we add explicit Gaussian noise to the input and decode $y$ from $e''_y = e'_y + \epsilon$ following (Nukrai et al., 2022; Zhou et al., 2022c). By introducing this noise, the uni-modal and multi-modal training processes become similar, and the learned decoder is more robust and invariant to the small perturbation $\mathcal{N}(0, \sigma^2 I)$, leading to improved performance.

Appendix Algorithm 1 summarizes the entire procedure of our proposed method, $C^3$, that enables learning cross-modal tasks with uni-modal data.

## 5 RESULTS

In this section, we verify the effectiveness of our proposed method, $C^3$, on four tasks: image captioning, audio captioning, video captioning, and text-to-image generation. We show that our method achieves state-of-the-art performances, generalizes to different modalities and contrastive embedding spaces, and is especially useful when multi-modal data are limited.

### 5.1 IMAGE CAPTIONING

We use the ClipCap model (Mokady et al., 2021), pairing a frozen CLIP ViT-B/32 image encoder (Radford et al., 2021) with a GPT-2 decoder (Radford et al., 2019). A lightweight MLP mapping network bridges the dimensional gap between CLIP (512-$d$) and GPT-2 (768-$d$) embeddings and also produces a prefix for GPT-2 caption generation. We train and evaluate on the MS-COCO dataset (Lin et al., 2014) using the standard split (Karpathy & Fei-Fei, 2015), comprising 113K training images and 5K each for validation and testing, with each image having 5 captions. We utilize metrics such as BLEU (Papineni et al., 2002) and ROUGE (Lin, 2004) to evaluate lexical and semantic similarity between generated and human captions.

We first train our model for text reconstruction using the MS-COCO captions only. Following $C^3$, we extract the text embedding from the frozen CLIP text encoder and apply the collapse operation (remove pre-computed mean) and corrupt operation (add Gaussian noise). After training, we evaluate our model in the cross-modal setting, replacing the text encoder with the CLIP image encoder and decoding captions from image embeddings. We refer to this evaluation setting as image-free zero-shot evaluation, as images are not seen during training. Additionally, we fine-tune the pre-trained model on different amounts of image-caption pairs and evaluate its performance. We refer to this evaluation setting as semi-supervised evaluation. More details can be found in Appendix E. We show image-free zero-shot captioning results in Table 2 and semi-supervised captioning results in Figure 6.

$C^3$ **achieves state-of-the-art image-free zero-shot captioning results.**  As shown in Table 2, our proposed method, $C^3$, outperforms previous state-of-the-art methods in image-free captioning. Details of the other methods can be found in Appendix Section A. Our ablation analysis demonstrates that both the collapse and corrupt components are crucial for improving cross-modal evaluation performance, as they eliminate the differences between embeddings from different modalities. Notably, the most competitive baseline, CapDec (Nukrai et al., 2022), can be viewed as an ablated version of $C^3$, but without an analysis of why the corruption works. Our proposed method, however, provides a clear explanation based on a geometric analysis of the multi-modal embedding space, and we further improve performance by introducing a collapse step. Overall, $C^3$ represents a potential standard approach for future works that use a multi-modal contrastive embedding space.

$C^3$ **is particularly useful in low-data regimes.**  In scenarios where multi-modal data is limited, our method remains highly effective. To demonstrate this, we fine-tuned our pre-trained model on 1%, 5%, 25%, and 100% of the MS-COCO training image-text pairs, and compared our performance with a fully supervised baseline (ClipCap) and ablated models (see Figure 6). The results clearly show that $C^3$ outperforms the fully supervised baseline across all metrics, with the most significant improvements seen in low-data regimes where multi-modal paired data are limited. Thus, our method represents a promising solution for achieving cross-modal tasks in such scenarios.

| Method | Conn. | Coll. | Corr. | BLEU-1↑ | BLEU-4↑ | METEOR↑ | ROUGE-L↑ | CIDEr↑ | SPICE↑ |
|---|---|---|---|---|---|---|---|---|---|
| Baselines | | | | | | | | | |
| ZeroCap (2022) | ✗ | ✗ | ✗ | 49.8 | 7.0 | 15.4 | 31.8 | 34.5 | - |
| MAGIC (2022) | ✗ | ✗ | ✗ | 56.8 | 12.9 | 17.4 | 39.9 | 49.3 | 11.3 |
| ESPER (2022) | ✗ | ✗ | ✗ | - | 21.9 | 21.9 | - | 78.2 | - |
| CLIPRe (2023) | ✓ | ✗ | ✗ | - | 4.6 | 13.3 | - | 25.6 | 9.2 |
| DeCap (2023) | ✓ | ✗ | ✗ | - | 8.9 | 17.5 | - | 50.6 | 13.1 |
| WS-ClipCap (2023) | ✓ | ✗ | ✗ | 50.3 | 9.6 | 15.2 | 37.5 | 33.7 | 8.6 |
| WS-ClipCap-Multi (2023) | ✓ | ✗ | ✓ | 65.5 | 22.1 | 22.2 | 48.0 | 74.6 | 14.9 |
| CapDec (2022) | ✓ | ✗ | ✓ | 69.2 | 26.4 | **25.1** | 51.8 | 91.8 | - |
| Ours | | | | | | | | | |
| $C^1$ | ✓ | ✗ | ✗ | 28.1 | 2.4 | 12.2 | 25.4 | 13.0 | 6.8 |
| $C_1^2$ | ✓ | ✓ | ✗ | 44.4 | 6.1 | 15.5 | 33.6 | 25.2 | 9.2 |
| $C_2^2$ | ✓ | ✗ | ✓ | 69.0 | 25.5 | 24.3 | 50.8 | 87.6 | 17.6 |
| $C^3$ | ✓ | ✓ | ✓ | $71.0_{\pm0.1}$ | $27.7_{\pm0.1}$ | $25.0_{\pm0.0}$ | $52.0_{\pm0.0}$ | $93.3_{\pm0.3}$ | $18.3_{\pm0.1}$ |

Table 2: Image-free image-to-text captioning results. We achieve state-of-the-art zero-shot image captioning and our ablation shows the effectiveness of each component in our method. Baseline results are from the original paper, where some metrics were not reported. Std with 3 runs reported.

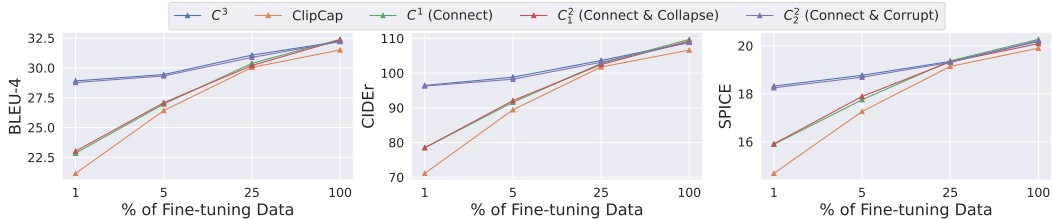

Figure 6: Image-to-text captioning results in the low data regime. When paired multi-modal data are limited, our approach that leverages uni-modal data for pre-training leads to substantial improvements compared to the purely supervised method (ClipCap).

**Qualitative analysis of collapse and corrupt.** Both the collapse and corrupt components of our method show consistent improvements, but it is not immediately clear how they do so. To address this, we provide qualitative results in Appendix E. We can see that after collapsing, the generated captions are much more natural and fluent, as it removes the most significant distributional difference between image and text embeddings. After corrupting, the model generates more accurate and faithful text descriptions of the image. We hypothesize that adding noise makes the decoder robust to small variations in the embedding space. Therefore when evaluating in cross-modal settings, the alignment noise will not affect the prediction and thus reduces hallucination in the generated caption.

## 5.2 TEXT-TO-IMAGE GENERATION

We further apply our method, $C^3$, to text-to-image generation, the reverse of image captioning. We utilize LAFITE (Zhou et al., 2022c), which integrates a frozen CLIP ViT-B/32 (Radford et al., 2021) as the text encoder and a modified StyleGAN2's generator (Karras et al., 2020) as the trainable decoder. We use the MS-COCO dataset (Lin et al., 2014) with LAFITE's official split, including 82K training and 40K validation images, with 5 captions each. We use the standard metrics such as FID (Fréchet Inception Distance) (Heusel et al., 2017) and IS (Inception Score) (Salimans et al., 2016) to assess the realism of generated images. Similar to the image captioning setting, we first train the model for image reconstruction using images only, and then evaluate the model in the cross-modal setting by generating images from text. We show the language-free zero-shot image generation results in Table 3. Similar to image captioning, we find

| Method | Conn. | Coll. | Corr. | FID↓ | IS↑ |
|---|---|---|---|---|---|
| Baselines | | | | | |
| DALL-E (2021) | ✗ | ✗ | ✗ | 27.5 | 17.9 |
| CogView (2021) | ✗ | ✗ | ✗ | 27.1 | 18.2 |
| LAFITE$_G$ (2022c) | ✓ | ✗ | ✓ | 20.9 | 24.9 |
| Ours | | | | | |
| $C^1$ | ✓ | ✗ | ✗ | 29.8 | 22.4 |
| $C_1^2$ | ✓ | ✓ | ✗ | 21.7 | 24.4 |
| $C_2^2$ | ✓ | ✗ | ✓ | 19.8 | 25.5 |
| $C^3$ | ✓ | ✓ | ✓ | **19.6** | **26.0** |

Table 3: Language-free text-to-image generation results. Our method $C^3$ consistently outperforms the baselines.

| | Image Captioning (MS-COCO 2014) | | | Audio Captioning (Clotho 2020) | | | Video Captioning (MSR-VTT 2016) | | |
|---|---|---|---|---|---|---|---|---|---|
| | BLEU-1$_\uparrow$ | METEOR$_\uparrow$ | ROUGE-L$_\uparrow$ | BLEU-1$_\uparrow$ | METEOR$_\uparrow$ | ROUGE-L$_\uparrow$ | BLEU-1$_\uparrow$ | METEOR$_\uparrow$ | ROUGE-L$_\uparrow$ |
| $C^1$ | 33.5 | 12.9 | 25.8 | 21.8 | 17.3 | 18.1 | 16.7 | 12.6 | 15.2 |
| $C_1^2$ | 53.8 | 17.4 | 38.6 | 26.7 | 20.0 | 21.4 | 25.1 | 17.8 | 23.1 |
| $C_2^2$ | 64.4 | 22.2 | 45.8 | 27.6 | 20.0 | 20.6 | 25.2 | 18.1 | 23.8 |
| $C^3$ | **74.0** | **26.6** | **54.0** | **29.5** | **20.1** | **23.0** | **31.4** | **20.0** | **26.9** |

Table 4: Generalization of $C^3$ to other modalities, datasets, and contrastive embedding spaces.

that our method $C^3$ **consistently outperforms the baselines** in terms of FID and IS. Our ablation study further reveals that each component of $C^3$ is useful for improving the performance. More detailed setups and qualitative comparisons can be found in Appendix F.

## 5.3 GENERALIZATION TO OTHER MODALITIES AND EMBEDDING SPACES

To verify the generalization of our method to other modalities, datasets, and embedding spaces, we further conduct experiments on zero-shot captioning from image, audio and video using ImageBind (Girdhar et al., 2023) embeddings. We use the same settings as image captioning with CLIP. Results are shown in Table 4. We find that $C^3$ **consistently improves baselines in all the settings**, and **using ImageBind embeddings achieves further improvements in image captioning compared to CLIP embeddings**.

## 6 RELATED WORKS (FULL VERSION IN APPENDIX A)

**Multi-modal contrastive learning and resulting geometry.** Multi-modal contrastive learning aims to bridge representations from different modalities, drawing similar concepts closer and distancing dissimilar ones (Radford et al., 2021). CLIP (Radford et al., 2021), ImageBind (Girdhar et al., 2023), and similar models have leveraged extensive multi-modal data to construct such representation spaces, which have been demonstrated to effectively support a range of uni-modal and multi-modal applications (Wortsman et al., 2022; Ramesh et al., 2022; Shen et al., 2022). However, the resulting geometry in the shared representation space, particularly the "modality gap", where embeddings from different modalities are clearly separate in the shared representation space, remains under-explored (Liang et al., 2022; Zhang et al., 2023). In our work, we unify the observations from Liang et al. (2022) and Zhang et al. (2023), and contribute a formal formulation and theoretical explanation of the unique geometry resulting from multi-modal contrastive learning.

**Learning cross-modal tasks with uni-modal data.** Given the expense of multi-modal data collection, there is a growing interest in learning cross-modal tasks using uni-modal data. Based on the assumption that contrastive optimization makes representations from different modalities interchangeable, recent works have leveraged these representation spaces and shown great success in building image captioning models with text data only (Tam et al., 2023; Li et al., 2023; Nukrai et al., 2022) and text-to-image generation models with image data only (Zhou et al., 2022c;a;b). Despite these advancements, these methods have noted the intriguing "modality gap" phenomenon and proposed different empirical methods to address this gap, such as (Tam et al., 2023)'s paraphrased decoding, (Li et al., 2023; Zhou et al., 2022a)'s memory retrieval, (Nukrai et al., 2022; Zhou et al., 2022c)'s noise addition, and (Zhou et al., 2022b)'s prior network. In our work, we first provide a theoretical analysis of the multi-modal representation space geometry. Based on the geometry, we propose a simple method that addresses the "modality gap" in a principled manner and ultimately improves performance on cross-modal tasks and outperforms these strategies.

## 7 CONCLUSION

In this work, we provide a theoretical explanation of the unique geometry that arises from multi-modal contrastive learning. Building upon this, we present a straightforward technique, $C^3$, which enhances the interchangeability of embeddings between modalities, enabling the creation of cross-modal applications using only uni-modal data. We demonstrate the effectiveness of our approach on image, audio, video captioning and text-to-image generation, achieving state-of-the-art performance on zero-shot evaluation settings when trained solely on uni-modal data.

## ACKNOWLEDGMENTS

We thank all the reviewers for their constructive feedback. Serena Yeung-Levy is a Chan Zuckerberg Biohub – San Francisco Investigator.

## ETHICS STATEMENT

Drawing from a deep understanding of the multi-modal contrastive representation space, our method enables the effortless creation of multi-modal content from uni-modal data. While this has the potential to revolutionize content generation, the risks associated with generation, such as creating harmful content, persists. We strongly emphasize the importance of ethically and responsibly employing any advancements built upon our approach, ensuring that their impact enhances the betterment of our digital ecosystem.

## REPRODUCIBILITY STATEMENT

We provide open-source implementation of our work at `https://github.com/yuhui-zh15/C3`. The implementations will enable researchers to reproduce all the experiments described here and run their own analyses on additional multi-modal models and datasets.

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

## A    RELATED WORKS

**Multi-modal contrastive learning and resulting geometry.**    Multi-modal contrastive learning aims to create a shared representation for different modalities by attracting similar while repelling dissimilar concepts from different modalities during the optimization process (Radford et al., 2021; Zhang et al., 2022; Xu et al., 2021; EleutherAI, 2021; Yuan et al., 2021; Girdhar et al., 2023). Recent works such as CLIP (Radford et al., 2021) have leveraged large-scale image-text data during pre-training, resulting in models that can build strong uni-modal and cross-modal applications. Connecting different modalities can be advantageous given the complementarity of different modalities. For example, connecting vision and language enables zero-shot categorization of visual objects (Radford et al., 2021; Yuan et al., 2021), explanation of model prediction errors or internal representations (Eyuboglu et al., 2022; Hernandez et al., 2022), diagnosis and rectification of vision models using language by composing different concepts (Zhang et al., 2023; Vendrow et al., 2023), and learning cross-modal tasks with uni-modal data.

However, the geometry resulting from multi-modal contrastive learning has received limited study. Recent work (Liang et al., 2022) found that there is a clear distinction between embeddings from different modalities in the shared representation space, which is referred to as the modality gap. Liang et al. (2022) correctly attributed the gap to the joint effect of model initialization and optimization. However, they did not study the geometric property of the gap and their theory cannot explain the geometry as well. Their theory can only show there is a distributional difference between image and text embeddings. Subsequently, Zhang et al. (2023) studied the geometric properties of the modality gap and found that the gap can be empirically well approximated by a constant vector orthogonal to the image or text embedding subspace. However, this work did not provide an explanation for how this unique geometry arises.

In our work, we unify the observations from Liang et al. (2022) and Zhang et al. (2023), and contribute a formal formulation and theoretical explanation of the unique geometry resulting from multi-modal contrastive learning. Specifically, we show two important factors of the geometry of the multi-modal representation space: modality gap and alignment noise, where the modality gap is a constant vector orthogonal to the image and text embedding span, and the alignment noise can be approximated by a Gaussian distribution. The modality gap arises due to the interplay between the dimensional collapse in model initialization and the resulting collapsed gradient during optimization, whereas the alignment noise is due to the stable region of contrastive loss. These explanations provide a deeper understanding of multi-modal contrastive learning and resulting geometry.

Moreover, this non-trivial geometry has important implications for building applications on top of a multi-modal representation space.

**Learning cross-modal tasks with uni-modal data.**   Multi-modal data are less abundant and more expensive to collect than uni-modal data, making it ideal to learn cross-modal tasks with uni-modal data. The recent rise of multi-modal contrastive learning provides this possibility, as similar concepts from different modalities establish close connections during the optimization process. Many recent works have leveraged CLIP, which aligns images and text to achieve image-to-text captioning with text-only data and text-to-image generation with image-only data. These methods assume and explore how to interchangeably use image embeddings and text embeddings resulting from multi-modal contrastive learning. They achieve better performance than prior methods that did not leverage a multi-modal representation space, outperforming ZeroCap (Tewel et al., 2022), MAGIC (Su et al., 2022), and ESPER (Yu et al., 2022) on captioning, and DALL-E (Ramesh et al., 2021) and CogView (Ding et al., 2021) on image generation. However, due to the peculiar geometry that arises from multi-modal contrastive learning, paired image embeddings and text embeddings are not collapsed to the same point, making it non-trivial to substitute one for the other. Therefore, researchers are proposing different methods to tackle this problem.

For image-to-text captioning, WS-ClipCap (Li et al., 2023) was the first work to leverage CLIP's text embeddings to decode text during training, finding that training to decode a paraphrased version of the text from the corresponding CLIP's text embedding leads to significantly better performance (WS-ClipCap-Multi). That is, for datasets where a single image is paired with multiple captions, it is better to decode a caption by feeding the text embedding obtained from one of the other captions corresponding to the same image. Despite proposing the method, WS-ClipCap failed to explain why it works and that on a high level, this paraphrased decoding can be viewed as adding noise. Decap (Li et al., 2023) maintains a memory of image embeddings and converts image embeddings to text embeddings by using a weighted average of the most similar image embeddings. They then feed the decoder with the converted embedding to decode the same text. While their method outperforms the baseline CLIPRe, which directly retrieves the most similar captions based on image embeddings, it underperforms WS-ClipCap-Multi, despite proposing a more complex method to tackle the modality gap. CapDec (Nukrai et al., 2022) found that directly adding Gaussian noise to the embedding and then feeding it into the decoder leads to comparably better performance. This method corresponds exactly to the corrupt stage in our method. CapDec attributes its success to the hypothesis that adding Gaussian noise closes the modality gap, however, this intuition is inaccurate.

For text-to-image generation, LAFITE (Zhou et al., 2022c) was the first work to use CLIP with uni-modal images only. Their approach can be seen as the inverse version of CapDec, where they add Gaussian noise to the image embedding and decode the same image. They also incorrectly suggest that adding Gaussian noise closes the modality gap. In contrast, DALLE-2 (Ramesh et al., 2022) uses a complex and heavily trained prior network to convert CLIP text embeddings to CLIP image embeddings but did not provide clear evidence of why the prior network is necessary and its effectiveness. Corgi (Zhou et al., 2022b) modifies the prior network by restricting its starting point, resulting in improved performance.

Despite the plethora of complex tricks proposed, there is no consensus on the best approach to interchangeably use image and text embeddings, and none of them reveal fundamental reasons or perform systematic ablations. Our work unifies previous approaches by first analyzing the multi-modal representation space and how it arises. Based on the geometric relation, we provide a straightforward method to tackle embedding interchangeability and demonstrate that our method achieves state-of-the-art results on cross-modal tasks when only training on uni-modal data.

## B  THEORY PROOF

In this section, we provide proofs of the two lemmas used in the main paper that reveal important properties of multi-modal contrastive learning.

## B.1 PROOF OF LEMMA 1

**Lemma 3. (Gradients in Contrastive Optimization)**
*With the mild assumption of equal presence of $n$ images and texts with $p(x_i) = p(y_i) = 1/n$, optimizing the multi-modal contrastive loss $\mathcal{L} = -\frac{1}{2n} \sum_{i=1}^{n} \big( \log \frac{\exp(e_{x_i} \cdot e_{y_i}/\tau)}{\sum_{j=1}^{n} \exp(e_{x_i} \cdot e_{y_j}/\tau)} + \log \frac{\exp(e_{x_i} \cdot e_{y_i}/\tau)}{\sum_{j=1}^{n} \exp(e_{x_j} \cdot e_{y_i}/\tau)} \big)$ yields the following gradients:*

$$\nabla_{e_{x_i}} \mathcal{L} = \lambda \sum_{j=1}^{n} \alpha_{y_j}(e_{y_j} - e_{y_i})$$

$$\nabla_{e_{y_i}} \mathcal{L} = \lambda \sum_{j=1}^{n} \alpha_{x_j}(e_{x_j} - e_{x_i})$$

*where $\lambda = 1/(2n\tau)$, $\alpha_{x_j} = p(x_j|y_i) + p(y_i|x_j)$, $\alpha_{y_j} = p(y_j|x_i) + p(x_i|y_j)$, $p(x_i|y_j) = \frac{\exp(e_{x_i} \cdot e_{y_j}/\tau)}{\sum_{k=1}^{n} \exp(e_{x_k} \cdot e_{y_j}/\tau)}$, $p(y_i|x_j) = \frac{\exp(e_{y_i} \cdot e_{x_j}/\tau)}{\sum_{k=1}^{n} \exp(e_{y_k} \cdot e_{x_j}/\tau)}$, and $\tau$ is temperature.*

*Proof of Lemma 1.* We first prove $\forall k, \sum_{i=1}^{n} p(x_k|y_i) = 1$ using Bayes' theorem:

$$\sum_{i=1}^{n} p(x_k|y_i) = \sum_{i=1}^{n} \frac{p(y_i|x_k)p(x_k)}{p(y_i)} = \sum_{i=1}^{n} \frac{p(y_i|x_k)(1/n)}{(1/n)} = 1$$

Then, we can prove the lemma using chain rule:

$$\nabla_{e_{x_k}} \mathcal{L}$$

$$= \nabla_{e_{x_k}} \big[ -\frac{1}{2n} \sum_{i=1}^{n} \big( \log \frac{\exp(e_{x_i} \cdot e_{y_i}/\tau)}{\sum_{j=1}^{n} \exp(e_{x_i} \cdot e_{y_j}/\tau)} + \log \frac{\exp(e_{x_i} \cdot e_{y_i}/\tau)}{\sum_{j=1}^{n} \exp(e_{x_j} \cdot e_{y_i}/\tau)} \big) \big]$$

$$= -\frac{1}{2n} \nabla_{e_{x_k}} \big[ \sum_{i=1}^{n} 2 e_{x_i} \cdot e_{y_i}/\tau - \sum_{i=1}^{n} \log(\sum_{j=1}^{n} \exp(e_{x_i} \cdot e_{y_j}/\tau)) - \sum_{i=1}^{n} \log(\sum_{j=1}^{n} \exp(e_{x_j} \cdot e_{y_i}/\tau)) \big]$$

$$= -\frac{1}{2n} \big[ 2 e_{y_k}/\tau - \nabla_{e_{x_k}} \log(\sum_{j=1}^{n} \exp(e_{x_k} \cdot e_{y_j}/\tau)) - \sum_{i=1}^{n} \nabla_{e_{x_k}} \log(\sum_{j=1}^{n} \exp(e_{x_j} \cdot e_{y_i}/\tau)) \big]$$

$$= -\frac{1}{2n} \big[ 2 e_{y_k}/\tau - \sum_{i=1}^{n} \frac{\exp(e_{x_k} \cdot e_{y_i}/\tau)}{\sum_{j=1}^{n} \exp(e_{x_k} \cdot e_{y_j}/\tau)} e_{y_i}/\tau - \sum_{i=1}^{n} \frac{\exp(e_{x_k} \cdot e_{y_i}/\tau)}{\sum_{j=1}^{n} \exp(e_{x_j} \cdot e_{y_i}/\tau)} e_{y_i}/\tau \big]$$

$$= -\frac{1}{2n\tau} \big[ 2 e_{y_k} - \sum_{i=1}^{n} p(y_i|x_k) e_{y_i} - \sum_{i=1}^{n} p(x_k|y_i) e_{y_i} \big]$$

$$= \frac{1}{2n\tau} \big[ \sum_{i=1}^{n} p(y_i|x_k)(e_{y_i} - e_{y_k}) + \sum_{i=1}^{n} p(x_k|y_i)(e_{y_i} - e_{y_k}) - (1 - \sum_{i=1}^{n} p(x_k|y_i)) e_{y_k} \big]$$

$$= \frac{1}{2n\tau} \big[ \sum_{i=1}^{n} (p(y_i|x_k) + p(x_k|y_i))(e_{y_i} - e_{y_k}) \big]$$

$$= \lambda \sum_{i=1}^{n} \alpha_{y_i}(e_{y_i} - e_{y_k})$$

Similarly, we can get $\nabla_{e_{y_k}} \mathcal{L} = \lambda \sum_{i=1}^{n} \alpha_{x_i}(e_{x_i} - e_{x_k})$, which finishes the proof. $\square$

## B.2 PROOF OF LEMMA 2

**Lemma 4. (Stable Region Controlled by Temperature)**
*We consider a single term in the multi-modal contrastive loss $\mathcal{L}_i = -\log \frac{\exp(e_{x_i} \cdot e_{y_i}/\tau)}{\sum_{j=1}^{n} \exp(e_{x_i} \cdot e_{y_j}/\tau)}$. We*

*define the margin $r = \boldsymbol{e}_{x_i} \cdot \boldsymbol{e}_{y_i} - \max_{j \neq i} \boldsymbol{e}_{x_i} \cdot \boldsymbol{e}_{y_j}$ as the measure of the similarity difference between the matched pair and the hardest negative pair. When $r$ exceeds a threshold given below, $\mathcal{L}_i$ falls below a small pre-set value $\delta$, where we assume optimization ends:*

$$r \geq \tau \log \frac{o(\tau)}{\exp(\delta) - 1},$$

*where $o(\tau)$ is a monotonically increasing function of temperature $\tau$ that satisfies $1 < o(\tau) < n$. Therefore, the required margin $r$ is monotonically increasing with $\tau$.*

*Proof of Lemma 2.* We first prove $\sum_i \exp(t_i/\tau) \leq o(\tau) \exp(\max_i t_i/\tau)$, where $o(\tau)$ is a monotonically increasing function of $\tau$ that satisfies $1 < o(\tau) < n$. Let us denote $m = \arg\max_i t_i$ (no tie), we have:

$$\sum_i \exp(t_i/\tau) = \exp(t_m/\tau)(1 + \sum_{i \neq m} \exp((t_i - t_m)/\tau))$$

Let us denote $o'(\tau) = 1 + \sum_{i \neq m} \exp((t_i - t_m)/\tau)$, since $\forall i \neq m, t_i - t_m < 0$, we have $0 < \exp((t_i - t_m)/\tau) < 1$, so $0 < \sum_{i \neq m} \exp((t_i - t_m)/\tau) < n - 1$, therefore $1 < o'(\tau) < n$. Moreover, $\exp((t_i - t_m)/\tau)$ monotonically increases with $\tau$, therefore $o'(\tau)$ is a monotonically increasing function of $\tau$. We can denote $o(\tau) = \lceil o'(\tau) \rceil$.

Based on this, we have:

$$
\begin{aligned}
\mathcal{L}_i \\
&= -\log \frac{\exp(\boldsymbol{e}_{x_i} \cdot \boldsymbol{e}_{y_i}/\tau)}{\sum_{j=1}^n \exp(\boldsymbol{e}_{x_i} \cdot \boldsymbol{e}_{y_j}/\tau)} \\
&= -\log \frac{1}{1 + \sum_{j \neq i} \exp((\boldsymbol{e}_{x_i} \cdot \boldsymbol{e}_{y_j} - \boldsymbol{e}_{x_i} \cdot \boldsymbol{e}_{y_i})/\tau)} \\
&= \log(1 + \sum_{j \neq i} \exp((\boldsymbol{e}_{x_i} \cdot \boldsymbol{e}_{y_j} - \boldsymbol{e}_{x_i} \cdot \boldsymbol{e}_{y_i})/\tau)) \\
&\leq \log(1 + o(\tau) \max_{j \neq i} \exp((\boldsymbol{e}_{x_i} \cdot \boldsymbol{e}_{y_j} - \boldsymbol{e}_{x_i} \cdot \boldsymbol{e}_{y_i})/\tau)) \\
&= \log(1 + o(\tau) \exp(-r/\tau))
\end{aligned}
$$

Suppose $\mathcal{L}_i \leq \log(1 + o(\tau) \exp(-r/\tau)) \leq \delta$, we have $r \geq \tau \log \frac{o(\tau)}{\exp(\delta)-1}$, which finishes the proof. $\qquad\square$

## C EMPIRICAL VERIFICATION OF GEOMETRY

In Section 3, we established a theoretical framework for the multi-modal contrastive representation space geometry. We prove that after contrastive learning, we have $\boldsymbol{e}_x - \boldsymbol{e}_y = \boldsymbol{c}_\perp + \boldsymbol{\epsilon}$, where $\boldsymbol{e}_x$ and $\boldsymbol{e}_y$ are $\ell_2$-normalized embeddings of a paired image $x$ and text $y$, $\boldsymbol{c}_\perp$ is a constant vector representing the underlined modality gap and is orthogonal to the image and text embedding span, and $\boldsymbol{\epsilon} \sim \mathcal{N}(\boldsymbol{0}, \sigma^2 \boldsymbol{I})$ is a random Gaussian vector representing the underlined alignment noise. Here we employ statistical methods to validate the proposed geometric structure on large pre-trained contrastive models. Figure 7 visualizes all the definitions introduced in this section.

To analyze the modality gap $\boldsymbol{c}_\perp$, we need to first find a way to isolate this vector, because it is entangled with alignment noise $\boldsymbol{\epsilon}$ in $\boldsymbol{e}_x - \boldsymbol{e}_y$. Since $\boldsymbol{\epsilon}$ is Gaussian, averaging multiple instances of $\boldsymbol{e}_x - \boldsymbol{e}_y$ should neutralize this noise, and thus isolating $\boldsymbol{c}_\perp$. Therefore, we randomly group every 100 image-text pairs into a group $i$. We define embedding difference for pair $j$ in group $i$ as $\boldsymbol{d}_j^{(i)} = \boldsymbol{e}_{x_j}^{(i)} - \boldsymbol{e}_{y_j}^{(i)}$. This difference includes both the modality gap and alignment noise. By computing the expected value, $\boldsymbol{d}^{(i)} = \mathbb{E}_j[\boldsymbol{d}_j^{(i)}]$, we effectively eliminate the noise component and leave the modality gap.

To demonstrate that the modality gap is a constant vector, we analyze its magnitude and direction across different groups. The distribution of $\|\boldsymbol{d}^{(i)}\|_2$ (Figure 8 (Gap Length)) shows that the gap has

a near constant length of 0.83, while the 0.99 mean of $\cos(\boldsymbol{d}^{(i)}, \boldsymbol{d}^{(j)})$ (Figure 8 (Gap Direction)) confirms that the gap has the same direction. These findings collectively validate that the modality gap is a constant vector.

Next, we establish the orthogonality of the modality gap to the embedding spans. By demonstrating its orthogonality to the image embedding span, we implicitly confirm its orthogonality to the text embedding span, given that the modality gap is a constant vector. We define the embedding difference of image $j$ and $k$ as $\boldsymbol{r}_{j,k}^{(i)} = \boldsymbol{e}_{x_j}^{(i)} - \boldsymbol{e}_{x_k}^{(i)}$. We observe the distribution of $\cos(\boldsymbol{d}^{(i)}, \boldsymbol{r}_{j,k}^{(i)})$ shown in Figure 8 (Gap Orthogonality) has zero mean with a small standard deviation 0.06, which demonstrates the gap's orthogonality to embedding spans.

Finally, we address the alignment noise. We define $\boldsymbol{\epsilon}_j^{(i)} = \boldsymbol{d}_j^{(i)} - \boldsymbol{d}^{(i)}$, which eliminates the gap and only leaves the noise. We first demonstrate its zero-mean nature $\mathbb{E}_{i,j}[\boldsymbol{\epsilon}_j^{(i)}] = \boldsymbol{0}$ through the distribution of each dimension's mean. We see the mean of each dimension of the noise vectors is bounded between -1e-8 and 1e-8 (Figure 8 (Noise Mean)). Furthermore, we show in Figure 8 (Noise Direction) that the distribution of $\cos(\boldsymbol{\epsilon}_j^{(i)}, \boldsymbol{\epsilon}_k^{(i)})$ has zero mean and small standard deviation 0.10, indicating that noise has random directions as every two noises are likely to be orthogonal. These findings affirm that the alignment noise can be approximated by a Gaussian noise $\mathcal{N}(\boldsymbol{0}, \sigma^2\boldsymbol{I})$.

In summary, our empirical analyses support the proposed geometric model of the multi-modal contrastive representation space, with a constant modality gap vector orthogonal to embedding spans and an alignment noise characterizable as zero-mean Gaussian.

In Figure 9, we show that our analyses above fully apply to other modalities, datasets, and contrastive embedding spaces, such as image-caption (MS-COCO), audio-caption (Clotho), and video-caption (MSR-VTT) on ImageBind embeddings.

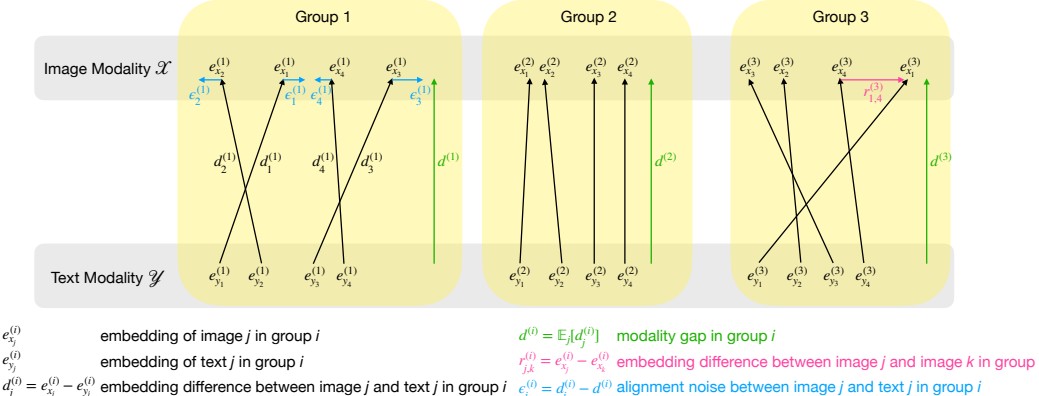

Figure 7: Visualization of the multi-modal contrastive representation space and various definitions introduced in Appendix C.

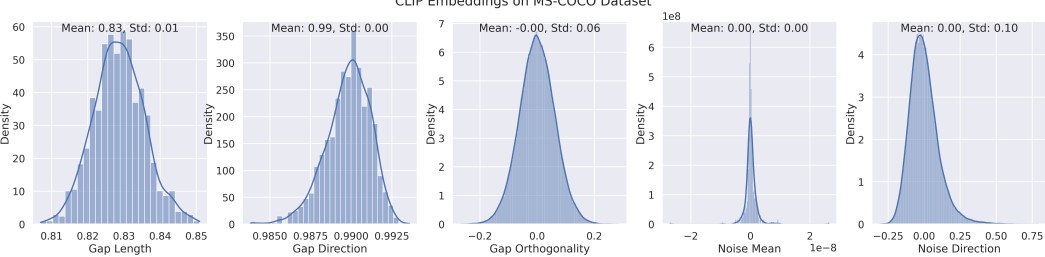

Figure 8: Empirical verification of the multi-modal contrastive representation space geometry. The modality gap approximates a constant vector, indicated by the gap length and direction distributions. The modality gap is orthogonal to the span of embeddings from two modalities, indicated by the gap orthogonality distributions. The alignment noise can be approximated by zero-mean Gaussian, indicated by the noise mean and direction distributions.

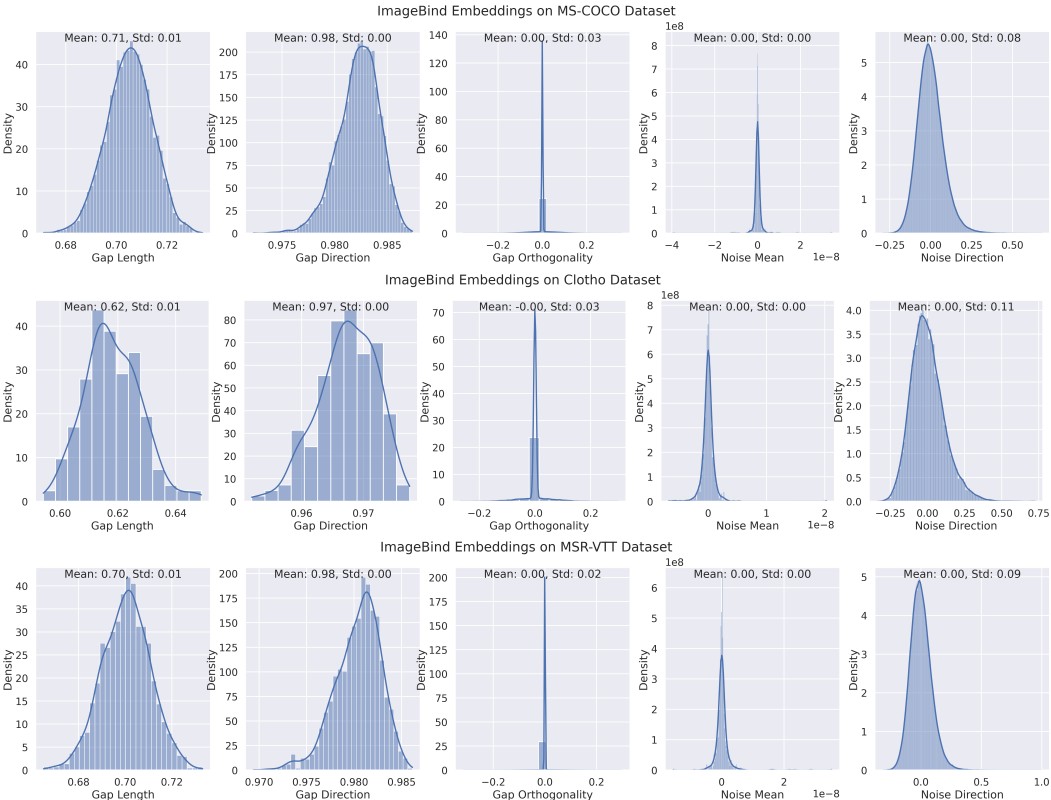

Figure 9: Generalization of Figure 8 to other modalities and multi-modal contrastive embedding spaces, including image-caption (MS-COCO), audio-caption (Clotho), and video-caption (MSR-VTT) on ImageBind embeddings.

# D  $C^3$ ALGORITHM

In this section, we summarize the $C^3$ algorithm as follows:

---

**Algorithm 1** $C^3$ algorithm

---

**Require:** unpaired uni-modal dataset $\mathcal{X}$, $\mathcal{Y}$, fixed encoders $f_{\mathcal{X}} : \mathcal{X} \mapsto \mathbb{R}^d$, $f_{\mathcal{Y}} : \mathcal{Y} \mapsto \mathbb{R}^d$ obtained from multi-modal contrastive learning, trainable decoder $g : \mathbb{R}^d \mapsto \mathcal{Y}$, noise level $\sigma$
1: $\bar{e}_x \leftarrow \sum_{x \in \mathcal{X}} \frac{1}{|\mathcal{X}|} f_{\mathcal{X}}(x)$
2: $\bar{e}_y \leftarrow \sum_{y \in \mathcal{Y}} \frac{1}{|\mathcal{Y}|} f_{\mathcal{Y}}(y)$
3:
4: **function** TRAIN($y, f_{\mathcal{Y}}, g$)
5: $\quad e_y \leftarrow f_{\mathcal{Y}}(y)$
6: $\quad \epsilon \leftarrow \mathcal{N}(\mathbf{0}, \sigma^2 \boldsymbol{I})$
7: $\quad \hat{y} \leftarrow g(e_y - \bar{e}_y + \epsilon)$
8: $\quad$ **return** $\mathcal{L}(\hat{y}, y)$
9: **end function**
10:
11: **function** TEST($x, f_{\mathcal{X}}, g$)
12: $\quad e_x \leftarrow f_{\mathcal{X}}(x)$
13: $\quad \hat{y} \leftarrow g(e_x - \bar{e}_x)$
14: $\quad$ **return** $\hat{y}$
15: **end function**

---

# E  IMAGE CAPTIONING

In this section, we provide additional experimental details and qualitative results of image captioning.

## E.1  EXPERIMENTAL SETUP

**Model.**  We employ the ClipCap model architecture (Mokady et al., 2021), which utilizes the CLIP ViT-B/32 (Radford et al., 2021) as the image encoder $f_{\mathcal{X}}$, and a mapping network with a pre-trained GPT-2 (Radford et al., 2019) as the decoder $g$. The mapping network is designed to handle the difference in embedding dimensions between CLIP (512-$d$) and GPT-2 (768-$d$) and to generate a "prefix" as input to GPT-2. It is implemented as a lightweight MLP with a single hidden layer, which transforms the CLIP embedding into prefix embeddings for GPT-2 to generate captions. During training, we fix the CLIP encoder to maintain the connection between image and text embeddings and only update the decoder, which includes the mapping network and GPT-2.

**Data.**  To train and evaluate our model, we use the MS-COCO image-caption dataset (Lin et al., 2014). We adopt the widely-used data split (Karpathy & Fei-Fei, 2015), which consists of a training set of approximately 113K images, and validation and test sets of 5K images each, where each image has 5 ground truth captions.

**Evaluation.**  We evaluate our model using various commonly-used image captioning metrics, including BLEU (Papineni et al., 2002), METEOR (Banerjee & Lavie, 2005), ROUGE (Lin, 2004), CIDEr (Vedantam et al., 2015), and SPICE (Anderson et al., 2016). These metrics measure the lexical and semantic similarities between the generated captions and the ground-truth captions.

**Setup.**  We train our model for text reconstruction using the MS-COCO captions only. Following the $C^3$ algorithm, we extract the text embedding from the CLIP text encoder $f_{\mathcal{Y}}$ and apply the collapse operation (removing the pre-computed mean) and corrupt operation (adding Gaussian noise). After pre-training, we evaluate our model in the cross-modal setting. We replace the encoder with the CLIP image encoder and decode captions from image embeddings. Since no image is seen during pre-training, we refer to this evaluation setting as image-free zero-shot evaluation. Additionally, we fine-tune our model on different amounts of image-caption pairs and evaluate its performance.

We refer to this evaluation setting as semi-supervised evaluation. During both the pre-training and fine-tuning stages, we train the model for 10 epochs with a batch size of 40, a learning rate of 2e-5, and AdamW (Loshchilov & Hutter, 2019) optimizer with a linear warmup of 5K steps. We use early stopping on the validation set and report the test set performance.

## E.2 QUALITATIVE EXAMPLES

We provide qualitative results for image captioning in Appendix Figure 10, which helps us better understand the improvements of each component of $C^3$. We observe that:

- $C^1$ generates captions that are **highly repetitive and/or nonsensical**.

- $C^2_1$ and $C^2_2$ generates captions that are **more fluent, but contain some hallucinations**.

- $C^3$ generates captions that are **more correct and concise with no extraneous details**.

## E.3 QUANTITATIVE RESULTS

| Method | Connect | Collapse | Corrupted | BLEU-1 | BLEU-4 | METEOR | ROUGE-L | CIDEr | SPICE |
|---|---|---|---|---|---|---|---|---|---|
| | | | | 1% Fine-tuning | | | | | |
| ClipCap | - | - | - | 64.5 | 21.2 | 21.2 | 47.6 | 71.1 | 14.7 |
| $C^1$ | ✓ | ✗ | ✗ | 66.5 | 22.9 | 22.6 | 48.7 | 78.5 | 15.9 |
| $C^2_1$ | ✓ | ✓ | ✗ | 66.8 | 23.0 | 22.6 | 48.7 | 78.5 | 15.9 |
| $C^2_2$ | ✓ | ✗ | ✓ | 71.7 | 28.8 | 25.3 | 52.7 | 96.3 | 18.3 |
| $C^3$ (Ours) | ✓ | ✓ | ✓ | 71.9 | 28.9 | 25.3 | 52.7 | 96.4 | 18.3 |
| | | | | 5% Fine-tuning | | | | | |
| ClipCap | - | - | - | 70.1 | 26.4 | 24.0 | 51.3 | 89.4 | 17.3 |
| $C^1$ | ✓ | ✗ | ✗ | 70.6 | 27.0 | 24.5 | 51.6 | 91.6 | 17.8 |
| $C^2_1$ | ✓ | ✓ | ✗ | 70.7 | 27.1 | 24.6 | 51.7 | 92.1 | 17.9 |
| $C^2_2$ | ✓ | ✗ | ✓ | 72.1 | 29.3 | 25.6 | 53.1 | 98.2 | 18.7 |
| $C^3$ (Ours) | ✓ | ✓ | ✓ | 72.3 | 29.4 | 25.7 | 53.1 | 98.8 | 18.8 |
| | | | | 25% Fine-tuning | | | | | |
| ClipCap | - | - | - | 73.0 | 30.0 | 26.0 | 53.7 | 101.7 | 19.1 |
| $C^1$ | ✓ | ✗ | ✗ | 73.2 | 30.4 | 26.1 | 53.9 | 102.8 | 19.4 |
| $C^2_1$ | ✓ | ✓ | ✗ | 73.1 | 30.2 | 26.1 | 53.8 | 102.4 | 19.3 |
| $C^2_2$ | ✓ | ✗ | ✓ | 73.2 | 30.9 | 26.3 | 54.2 | 103.1 | 19.3 |
| $C^3$ (Ours) | ✓ | ✓ | ✓ | 73.4 | 31.1 | 26.3 | 54.3 | 103.6 | 19.4 |
| | | | | 100% Fine-tuning | | | | | |
| ClipCap | - | - | - | 74.0 | 31.5 | 26.8 | 54.7 | 106.6 | 19.9 |
| $C^1$ | ✓ | ✗ | ✗ | 74.6 | 32.4 | 27.2 | 55.2 | 109.7 | 20.3 |
| $C^2_1$ | ✓ | ✓ | ✗ | 74.6 | 32.3 | 27.1 | 55.2 | 109.2 | 20.1 |
| $C^2_2$ | ✓ | ✗ | ✓ | 74.1 | 32.2 | 27.1 | 55.2 | 108.8 | 20.2 |
| $C^3$ (Ours) | ✓ | ✓ | ✓ | 74.0 | 32.2 | 27.1 | 55.2 | 108.9 | 20.2 |

Table 5: Image-to-text captioning results in the low data regime. When paired multi-modal data are limited, our approach that leverages uni-modal data for pre-training leads to substantial improvements compared to the purely supervised method (ClipCap). This is the table version used to reproduce Figure 5 in the main paper. Results are averaged over three random seeds for 1-25% fine-tuning to reduce the effect of randomness.

We include Appendix Table 5, which is the table version to produce Figure 5 in the main text. In this table, we compare the fully supervised ClipCap (Mokady et al., 2021) to an ablation of components in our method $C^3$, and demonstrate the effectiveness of learning cross-modal tasks with uni-modal data. Results are averaged over three random seeds for 1-25% fine-tuning to reduce the effect of randomness.

## F TEXT-TO-IMAGE GENERATION

In this section, we provide additional experimental details and qualitative results of text-to-image generation.

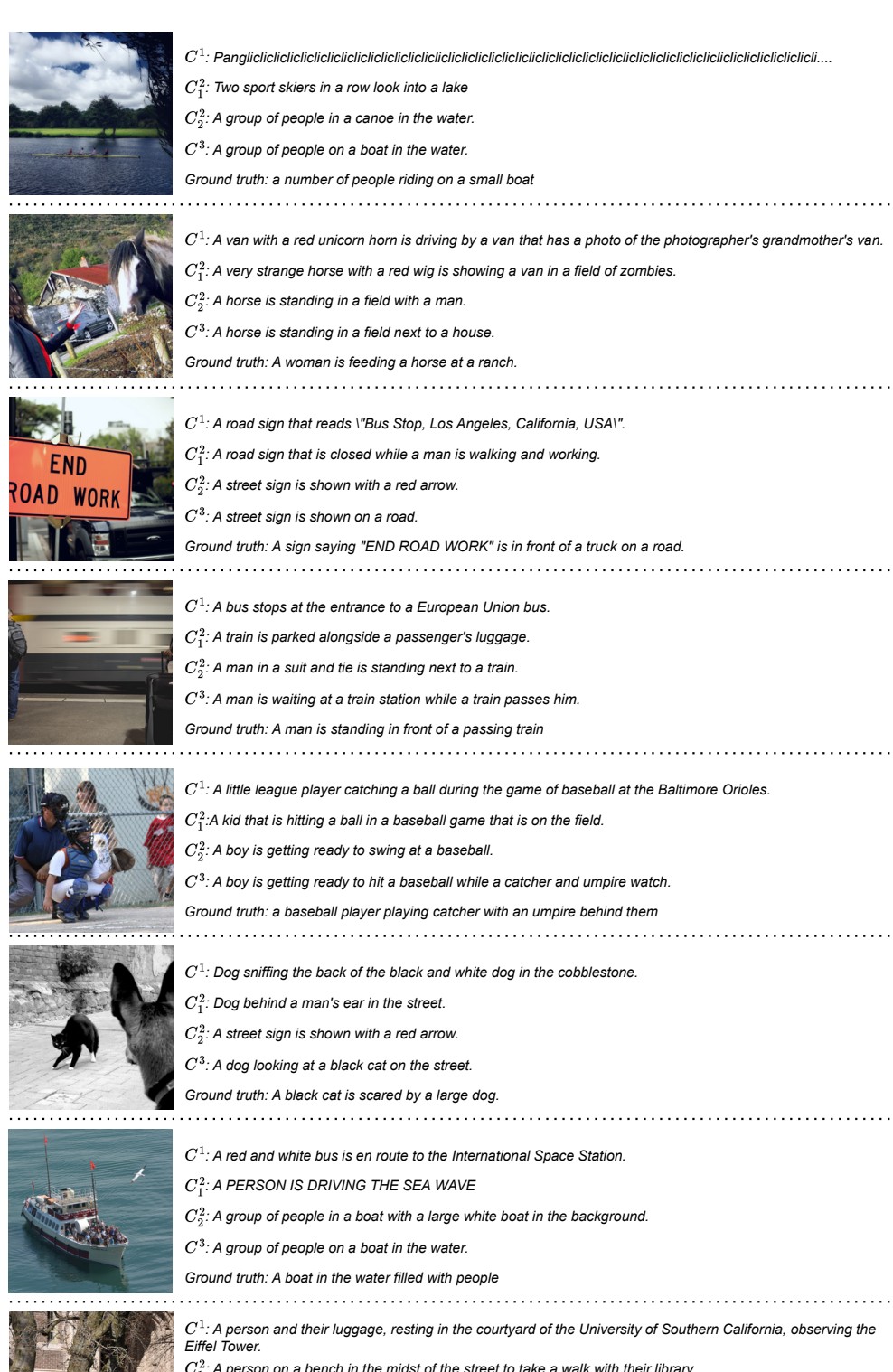

$C^1$: Pangliclicliclicliclicliclicliclicliclicliclicliclicliclicliclicliclicliclicliclicliclicliclicliclicliclicliclicliclicliclicliclicli....

$C^2_1$: Two sport skiers in a row look into a lake

$C^2_2$: A group of people in a canoe in the water.

$C^3$: A group of people on a boat in the water.

Ground truth: a number of people riding on a small boat

. . . . . . . . . . . . . . . . . . . . . . . . . . . . . . . . . . . . . . . . . . . . . . . . . . . . . . . . . . . . . . . . . . . . . . . . . . . . . . . . . . . . .

$C^1$: A van with a red unicorn horn is driving by a van that has a photo of the photographer's grandmother's van.

$C^2_1$: A very strange horse with a red wig is showing a van in a field of zombies.

$C^2_2$: A horse is standing in a field with a man.

$C^3$: A horse is standing in a field next to a house.

Ground truth: A woman is feeding a horse at a ranch.

. . . . . . . . . . . . . . . . . . . . . . . . . . . . . . . . . . . . . . . . . . . . . . . . . . . . . . . . . . . . . . . . . . . . . . . . . . . . . . . . . . . . .

$C^1$: A road sign that reads \"Bus Stop, Los Angeles, California, USA\".

$C^2_1$: A road sign that is closed while a man is walking and working.

$C^2_2$: A street sign is shown with a red arrow.

$C^3$: A street sign is shown on a road.

Ground truth: A sign saying "END ROAD WORK" is in front of a truck on a road.

. . . . . . . . . . . . . . . . . . . . . . . . . . . . . . . . . . . . . . . . . . . . . . . . . . . . . . . . . . . . . . . . . . . . . . . . . . . . . . . . . . . . .

$C^1$: A bus stops at the entrance to a European Union bus.

$C^2_1$: A train is parked alongside a passenger's luggage.

$C^2_2$: A man in a suit and tie is standing next to a train.

$C^3$: A man is waiting at a train station while a train passes him.

Ground truth: A man is standing in front of a passing train

. . . . . . . . . . . . . . . . . . . . . . . . . . . . . . . . . . . . . . . . . . . . . . . . . . . . . . . . . . . . . . . . . . . . . . . . . . . . . . . . . . . . .

$C^1$: A little league player catching a ball during the game of baseball at the Baltimore Orioles.

$C^2_1$: A kid that is hitting a ball in a baseball game that is on the field.

$C^2_2$: A boy is getting ready to swing at a baseball.

$C^3$: A boy is getting ready to hit a baseball while a catcher and umpire watch.

Ground truth: a baseball player playing catcher with an umpire behind them

. . . . . . . . . . . . . . . . . . . . . . . . . . . . . . . . . . . . . . . . . . . . . . . . . . . . . . . . . . . . . . . . . . . . . . . . . . . . . . . . . . . . .

$C^1$: Dog sniffing the back of the black and white dog in the cobblestone.

$C^2_1$: Dog behind a man's ear in the street.

$C^2_2$: A street sign is shown with a red arrow.

$C^3$: A dog looking at a black cat on the street.

Ground truth: A black cat is scared by a large dog.

. . . . . . . . . . . . . . . . . . . . . . . . . . . . . . . . . . . . . . . . . . . . . . . . . . . . . . . . . . . . . . . . . . . . . . . . . . . . . . . . . . . . .

$C^1$: A red and white bus is en route to the International Space Station.

$C^2_1$: A PERSON IS DRIVING THE SEA WAVE

$C^2_2$: A group of people in a boat with a large white boat in the background.

$C^3$: A group of people on a boat in the water.

Ground truth: A boat in the water filled with people

. . . . . . . . . . . . . . . . . . . . . . . . . . . . . . . . . . . . . . . . . . . . . . . . . . . . . . . . . . . . . . . . . . . . . . . . . . . . . . . . . . . . .

$C^1$: A person and their luggage, resting in the courtyard of the University of Southern California, observing the Eiffel Tower.

$C^2_1$: A person on a bench in the midst of the street to take a walk with their library.

$C^2_2$: A man sitting on a bench with a book.

$C^3$: A man and woman sitting on a park bench.

Ground truth: A man and woman sitting on a park bench under a huge tree

Figure 10: Qualitative examples of image-to-text captioning on the MS-COCO dataset. $C^1$ generates captions that are highly repetitive and/or nonsensical. $C^2_1$ and $C^2_2$ generates captions that are more fluent, but contain some hallucinations. $C^3$ generates captions that are more correct and concise with no extraneous details.

### F.1 Experimental Setup

**Model.** We use the LAFITE (Zhou et al., 2022c) model for image generation, which uses the CLIP ViT-B/32 (Radford et al., 2021) as the text encoder $f_{\mathcal{X}}$, and an adapted version of the unconditional generator from StyleGAN2 (Karras et al., 2020) as the decoder $g$. LAFITE's generator is adversarially trained alongside a discriminator with an additional contrastive objective to align the generator's representation space to that of CLIP's. As with image captioning, during training, we fix the CLIP encoder to maintain the connection between image and text embeddings, and only update the decoder, which in this case, involves updating both the generator and discriminator.

**Data.** Same as image-to-text captioning, we train and evaluate the model on the MS-COCO dataset (Lin et al., 2014). We use the same pre-processed data and data split provided in the LAFITE official code repository, comprised of 82K training images and 40K validation images with 5 captions per image.

**Evaluation.** We evaluate our model using the widely-used image generation metric Frechet Inception Distance (FID) (Heusel et al., 2017). This metric measure the realism of generated images by computing feature similarity between those of generated images and those of ground-truth images, where the features are derived from the pre-trained Inception-v3 model (Szegedy et al., 2016). We also report Inception Score (IS) (Salimans et al., 2016), which is similar to FID. Similarly to LAFITE, FID/IS scores are computed based on 50K generated images, using captions that are randomly sampled from the validation set.

**Setup.** We train the model for at most 750,000 steps with a batch size of 16, a learning rate of 2.5e-3, and Adam optimizer. We initialize our model with the official pre-trained weights from LAFITE (Zhou et al., 2022c) that was trained on CC3M (Sharma et al., 2018). We report the validation set performance with the lowest FID score.

### F.2 Qualitative Examples

We provide qualitative results for image generation in Figure 11, which helps us better understand the improvements of each component of $C^3$. We observe that:

- $C^1$ generates the **basic scene** that pertains to the text description, but are less photo-realistic in terms of contrast and color.
- $C_1^2$ and $C_2^2$ add more **fine-grained detail** despite still generating some artifacts.
- $C^3$ generates **sharper images** that are more **detailed with fewer artifacts**.

## G    Why Align Embedding from Different Modalities?

If embeddings from different modalities are aligned, we can train a model on one modality and then infer on another modality, enabling us to build cross-modal applications with only uni-modal data. This is an emerging field that lacks principled approaches to be easily applied without requiring more empirical tuning.

We added an experiment to explain this further. We train a text generator (image captioner) over CLIP's text embeddings $x$. During inference, we manually shift all the $x$ to $x + c$ to simulate the modality gap (a constant vector orthogonal to original spans). We report the captioning performance in terms of gap distance $\|c\|$ in Table 6. We observe substantial performance drops when the gap grows, showing the need to align embeddings.

## H    Effectiveness of Collapse vs Corrupt

In Table 2, we observe that adding noise (i.e., corrupt, $C_2^2$) is more effective than closing the modality gap (i.e., collapse, $C_2^1$). We hypothesize that the greater effectiveness of $C_2^2$ is because $C_2^2$ has two effects: 1) injecting noise in the span to mitigate alignment noise; 2) injecting noise in the modality gap direction to mitigate the model's sensitivity to this gap.

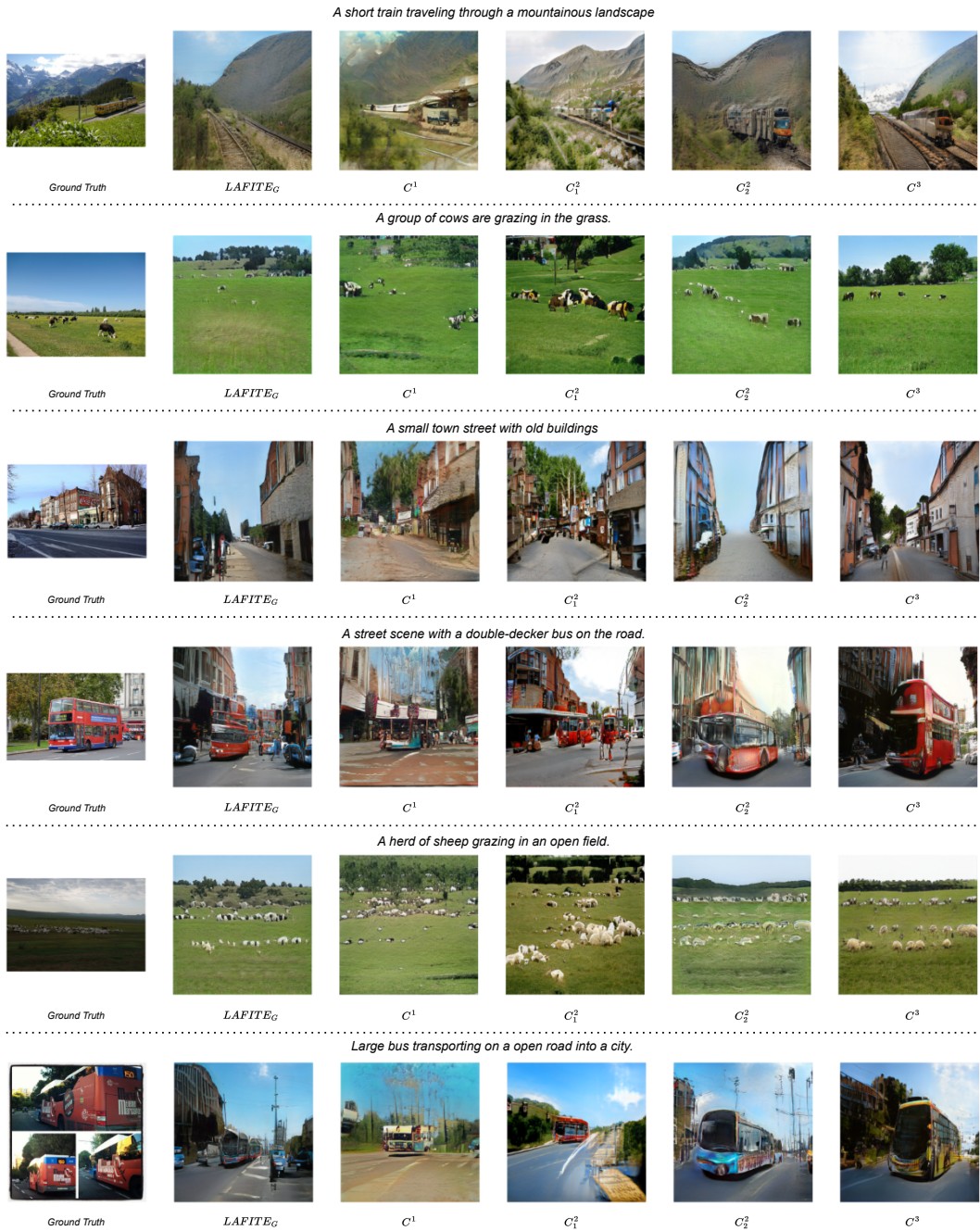

Figure 11: Qualitative examples of text-to-image generation on the MS-COCO dataset. $C^1$ generates the basic scene that pertains to the text description, but are less photo-realistic in terms of contrast and color. $C_1^2$ and $C_2^2$ add more fine-grained detail despite still generating some artifacts. $C^3$ generates sharper images that are more detailed with fewer artifacts.

| Gap Distance $\|c\|$ | ROUGE-1 | ROUGE-L | METEOR |
|---|---|---|---|
| 0.0 | 85.5 | 81.2 | 83.1 |
| 0.2 | 76.3 | 71.5 | 73.8 |
| 0.4 | 55.8 | 50.6 | 52.5 |
| 0.6 | 40.6 | 35.9 | 36.6 |
| 0.8 | 30.5 | 26.7 | 26.5 |
| 1.0 | 24.1 | 21.0 | 20.3 |
| 1.5 | 16.6 | 14.4 | 13.8 |
| 2.0 | 13.8 | 12.1 | 11.6 |

Table 6: Image captioning performance when trained on embedding $x$ and tested on $x + c$. This shows the necessity to align embeddings when training a model on one modality and then inferring on another modality.

We have added an experiment to verify this hypothesis. When adding noise sampled from Gaussian distributions, we remove its component in the modality gap direction. Specifically, given $\epsilon \sim \mathcal{N}(0, \sigma^2 I)$, we compute its projection on the gap direction as $\epsilon_g = \frac{\epsilon \cdot g}{\|g\|} \frac{g}{\|g\|}$, where $\frac{g}{\|g\|}$ is the modality gap direction, then we remove this projection to get a new noise $\epsilon' = \epsilon - \epsilon_g$. We add this new noise during training and name this experiment as $C_2^2$ (span noise only).

From Table 7, we see that adding noise only in the span (i.e., $C_2^2$ (span noise only)) makes its performance much worse than adding noise to all the directions (i.e., $C_2^2$), and its performance is similar to removing the modality gap (i.e., $C_1^2$). Therefore, adding noise (i.e., corrupt) actually leads to a similar improvement to removing the modality gap (i.e., collapse). The reason for the greater effectiveness of corrupt than collapse is that injecting Gaussian noise not only adds noise in the span but also to the modality gap direction.

Given the substantial size of the modality gap, adding noise is not enough to fully diminish the gap. Therefore, adding noise and removing the gap (i.e., $C^3$) still enhance the overall performance.

| Method | Conn. | Coll. | Corr. | BLEU-1$_\uparrow$ | BLEU-4$_\uparrow$ | METEOR$_\uparrow$ | ROUGE-L$_\uparrow$ | CIDEr$_\uparrow$ | SPICE$_\uparrow$ |
|---|---|---|---|---|---|---|---|---|---|
| $C^1$ | ✓ | ✗ | ✗ | 28.1 | 2.4 | 12.2 | 25.4 | 13.0 | 6.8 |
| $C_1^2$ | ✓ | ✓ | ✗ | 44.4 | 6.1 | 15.5 | 33.6 | 25.2 | 9.2 |
| $C_2^2$ (span noise only) | ✓ | ✗ | ✓ | 41.2 | 6.2 | 14.9 | 33.6 | 22.8 | 8.3 |
| $C_2^2$ | ✓ | ✗ | ✓ | 69.0 | 25.5 | 24.3 | 50.8 | 87.6 | 17.6 |
| $C^3$ | ✓ | ✓ | ✓ | **71.0** | **27.7** | 25.0 | **52.0** | **93.3** | **18.3** |

Table 7: Collapse is actually as effective as corrupt. The reason for the greater effectiveness of corrupt than collapse is that injecting Gaussian noise not only adds noise in the span but also to the modality gap direction.

## I  DIMENSIONAL COLLAPSE

Echoing our main paper, the phenomenon of dimensional collapse (Jing et al., 2022) in randomly initialized image and text encoders creates a modality gap prior to optimization that also persists after optimization. In this section, we offer further insights into dimensional collapse and demonstrate that 1) it is an inherent characteristic of deep neural networks and 2) deeper networks experience more pronounced dimensional collapse. Additionally, we establish a connection between dimensional collapse and the cone effect identified by Liang et al. (2022).

### I.1  REAL NETWORKS

In Figure 12, we find that the vision encoder (vision transformer) and text encoder (transformer) of the CLIP have the dimensional collapse phenomenon at initialization. Specifically, the effective dimension of the embeddings generated by the vision and text encoder is both much smaller than the full dimension, which induces a modality gap. Since the gradients of multi-modal contrastive

learning will only be propagated in the effective dimensions, the modality gap will be preserved after optimization.

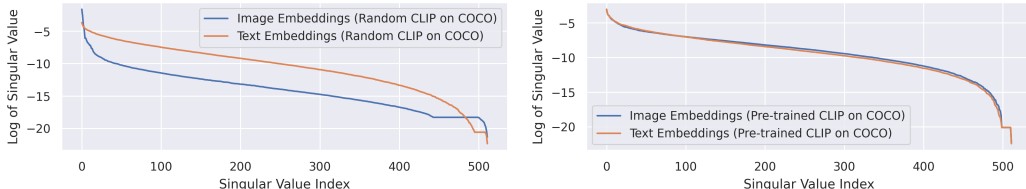

Figure 12: Dimensional collapse of the randomly initialized (left) and pre-trained (right) CLIP representation space. Singular values obtained from SVD reveal that the effective dimension of the image and text representation space is much smaller than the total number of dimensions.

## I.2 SIMULATION

To investigate the dimensional collapse phenomenon, we initialize a simple Multi-Layer Perceptron (MLP) with $n$ blocks, where each block contains a linear layer and Rectified Linear Unit (ReLU) activation. We set the dimensionality of the input space and hidden states to 512, and initialize the weights with Xavier uniform distribution and biases to zero. We initialize $N = 1000$ input embeddings, where each dimension is sampled from a standard normal distribution $\mathcal{N}(0, 1)$. We feed these embeddings into the MLP and extract the features after every 5 layers. We perform Singular Value Decomposition (SVD) on the feature covariance matrix to quantify the degree of dimensional collapse.

As shown in Figure 13, our experiments reveal two key insights:

1. Dimensional collapse is an inherent characteristic of deep neural networks and even a simple MLP exhibits this behavior.

2. Deeper networks are more prone to dimensional collapse, resulting in a smaller effective dimension of the feature space.

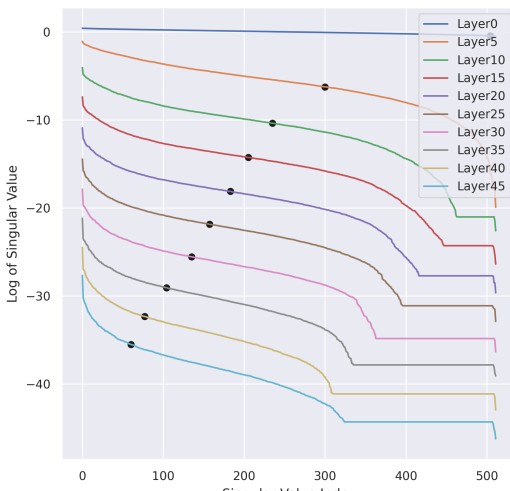

Figure 13: Simulation of dimensional collapse on a MLP ($n$*(ReLU(Linear))) network. The $x$-coordinates of the black dots indicate the effective dimensions for the embeddings from each of the layers, respectively, which quantifies the extent of collapse.

### I.3 CONNECTION TO CONE EFFECT

The dimensional collapse phenomenon can provide an explanation for the cone effect observed by Liang et al. (2022). They found that the cosine similarities between any two embeddings outputted by a deep neural network were significantly higher than zero. Due to the dimensional collapse, all embeddings share similar values in the ineffective dimensions, leading to high cosine similarities. The stronger the dimensional collapse, the greater the number of ineffective dimensions, and hence, a stronger cone effect. This explanation is illustrated in Figure 14, where collapsing the $z$-axis of a 3D representation space induces a cone, leading to significantly higher cosine similarities between any two embeddings.

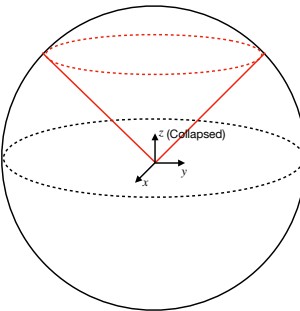

Figure 14: Dimensional collapse explains the cone effect of deep neural networks. When the $z$-axis of a 3D representation space is collapsed, it results in a cone shape, where the cosine similarities between any two embeddings are significantly higher than zero.

