# OpenReview forum: "Connect, Collapse, Corrupt: Learning Cross-Modal Tasks with Uni-Modal Data"
_ICLR.cc/2024/Conference — ICLR 2024 poster_

### Official Review · Reviewer_XDw2 · 2023-10-28

**Soundness:** 2 fair
**Presentation:** 3 good
**Contribution:** 2 fair
**Rating:** 6
**Confidence:** 4

**Summary:**

The paper studies the space's geometry of the features learned by contrastive learning and finds there is a modality gap in such feature space. To bridge the modality gap, based on the analysis of the geometry, they propose a three-step method called C^3 (Connect, Collapse, Corrupt). They conduct experiments on zero-shot image/audio/video captioning and text-to-image generation.

**Strengths:**

1. The discussion about the gradient direction in contrastive optimization in Lemma 1 sheds light on how the features learned by contrastive learning are processed.
2. The theoretical analysis and experiment results of the alignment noise are reasonable and effective.

**Weaknesses:**

1. The abstract and introduction emphasize the importance of the modality gap of the feature space,
however, the results in the experiments seem show that the key component is the alignment noise.
For instance, in Table 2, C_2^2 (only use connect and corrupt) can achieve a similar performance as the C^3.
Compared to C^1, only adding collapse boost 16.3 but 41 when only adding alignment noise.

2. In section 3.3, it provides some statistic values to show the space geometry.
I think such statistics may not actually reveal the space geometry since it averages all possible values.
Is that better to provide a histogram of such statistics to demonstrate the space geometry?
Furthermore, based on the constant modality gap analysis, is that the E_{i,j}[cos(d_i,d_j)] should have a value close to 1 since they should be parallel except for the noise effect.

3. The experiments are conducted on generation tasks, the quantitative performance is similar for the method and baseline,
while in the qualitative examples, I can not tell which method is better based on three examples. It will be better to provide more qualitative examples.

**Questions:**

1. In the analysis of the space geometry of contrastive features, the paper proposes that the modality gap is a constant vector and orthogonal to each modality. But it can not be derived easily that why the modality gap is the constant vector from the dimension collapse as mentioned in section 3.1. Are there any formal propositions for the constant modality gap vector in the initialization stage?

2. The paper aims to mitigate the modality gap between feature space of different modalities, but in the generation task or encoder/decoder-based architecture, is that the closeness of the features from different modalities indicate a better generation performance? I think it would be better to refer to previous works or conduct this kind of experiment to show the assumption of this paper is true in real applications.

---

> ### Author Response · Authors · 2023-11-19
>
> We thank Reviewer XDw2 for reviewing our paper and providing helpful feedback on our work. We address Reviewer XDw2’s concerns below.
>
> **Histogram of statistics revealing the space geometry**
>
> > In section 3.3, it provides some statistic values to show the space geometry. I think such statistics may not actually reveal the space geometry since it averages all possible values. Is that better to provide a histogram of such statistics to demonstrate the space geometry? Furthermore, based on the constant modality gap analysis, is that the $E_{i,j}[\cos(d_i,d_j)]$ should have a value close to 1 since they should be parallel except for the noise effect.
>
> Thank you for the suggestion and question. We have now included the histogram, as well as a detailed explanation of how to interpret these statistics in Appendix C.
>
> For $E_{i,j}[\cos(d_i,d_j)]$, because there is a noise entangled in $d_i = c_\perp + \epsilon_i$ and $d_j = c_\perp + \epsilon_j$, the cosine similarity will not be 1. However, if we compute the group-level $E_{i,j}[\cos(d_{c_i},d_{c_j})]$, where $c_i$ and $c_j$ are 100 randomly sampled images, the value reaches 0.99. We provide a detailed explanation in Appendix C.
>
> **Why corrupt ($C_2^2$) seems more effective than collapse ($C_1^2$)?**
>
> > The abstract and introduction emphasize the importance of the modality gap of the feature space, however, the results in the experiments seem show that the key component is the alignment noise. For instance, in Table 2, $C_2^2$ (only use connect and corrupt) can achieve a similar performance as the $C^3$. Compared to $C^1$, only adding collapse boost 16.3 but 41 when only adding alignment noise.
> >
>
> Thanks for the question. While $C_2^2$ (corrupt) seems to be more effective than $C_1^2$ (collapse) in Table 2 (image captioning on CLIP embedding space), they seem to be similarly effective in Table 3 (text-to-image generation on CLIP embedding space) and Table 4 (image / video / audio captioning on ImageBind embedding space).
>
> We hypothesize that the greater effectiveness of $C_2^2$ is because $C_2^2$ injects random noise in the embedding space during training, including the direction of the modality gap, which potentially reduces the model's sensitivity to this gap. Nevertheless, given the substantial size of the modality gap, $C_1^2$ remains necessary to diminish this gap, thereby enhancing the overall performance of $C^3$.
>
> It is also worth noting that $C^3$ adds near-zero additional computational cost over baselines. As our experiments demonstrate its effectiveness across a wide range of tasks, it can be a template for future work.
>
> **Significance of the quantitative improvement and more qualitative examples**
>
> > The experiments are conducted on generation tasks, the quantitative performance is similar for the method and baseline, while in the qualitative examples, I can not tell which method is better based on three examples. It will be better to provide more qualitative examples.
> >
>
> Thank you for the suggestion. We have added more qualitative examples for image captioning and text-to-image generation in Appendix Figure 10 and Figure 11. We hope these examples provide additional evidence of our method’s effectiveness.
>
> We also emphasize that the improvement of our method is significant. Per Reviewer bNKP’s suggestion, we have run $C^3$ in Table 2 three times with random seeds 1234, 5678, and 910 and reported the numbers below. There is a very small variance across different runs, and $C^3$ consistently outperforms all the baselines.
>
> | Seed | BLEU-1 | BLEU-4 | METEOR | ROUGE-L | CIDER | SPICE |
> | --- | --- | --- | --- | --- | --- | --- |
> | 1234 | 71.0 | 27.6 | 25.0 | 52.0 | 93.2 | 18.3 |
> | 5678 | 71.1 | 27.6 | 24.9 | 52.0 | 93.6 | 18.4 |
> | 910 | 70.9 | 27.8 | 25.0 | 52.1 | 93.0 | 18.2 |
> | Avg | 71.0 | 27.7 | 25.0 | 52.0 | 93.3 | 18.3 |
> | Std | 0.1 | 0.1 | 0.0 | 0.0 | 0.3 | 0.1 |

---

> > ### Author Response · Authors · 2023-11-19
> >
> > **Why modality gap vector is constant?**
> >
> > > In the analysis of the space geometry of contrastive features, the paper proposes that the modality gap is a constant vector and orthogonal to each modality. But it can not be derived easily that why the modality gap is the constant vector from the dimension collapse as mentioned in section 3.1. Are there any formal propositions for the constant modality gap vector in the initialization stage?
> > >
> >
> > Thanks for the question. The constant vector of the modality gap *does not result from the initialization stage only*, but results from the joint effect of model initialization and optimization:
> >
> > - During initialization, due to the dimensional collapse phenomenon (Definition 1), some dimensions of image and text embeddings are near constant, which we term ineffective dimensions.
> > - During optimization, the gradients will never be propagated into the shared ineffective dimensions between image and text embeddings (Lemma 1), causing these dimensions to remain the same constant.
> > - Finally, effective dimensions are aligned, and ineffective dimensions remain at initial values, resulting in a constant modality gap orthogonal to each embedding span.
> >
> > Here we provide an example to help understand the process:
> >
> > Suppose there are two image-text pairs in 3D space; only the 1st dimension of the image embeddings is effective, only the 2nd dimension of the text embeddings is effective, and all the other dimensions are ineffective.
> >
> > During initialization, let us denote these embeddings as $x_1, x_2, y_1, y_2 \in \mathbb{R}^3$:
> >
> > $x_1 = [0.7, 0.5, -0.5]$ (the 2nd, 3rd dimensions are constant across $x_i$)
> >
> > $x_2 = [-0.7, 0.5, -0.5]$ (the 2nd, 3rd dimensions are constant across $x_i$)
> >
> > $y_1 = [0.5, 0.7, 0.5]$ (the 1st, 3rd dimensions are constant across $y_i$)
> >
> > $y_2 = [0.5, -0.7, 0.5]$ (the 1st, 3rd dimensions are constant across $y_i$)
> >
> > During optimization, only the union of effective dimensions will be aligned, and the intersection of ineffective dimensions will remain constant:
> >
> > $x_1 = [0.6, 0.6, -0.5]$ (the 3rd dimension remains constant because of no gradient)
> >
> > $x_2 = [-0.6, -0.6, -0.5]$ (the 3rd dimension remains constant because of no gradient)
> >
> > $y_1 = [0.6, 0.6, 0.5]$ (the 3rd dimension remains constant because of no gradient)
> >
> > $y_2 = [-0.6, -0.6, 0.5]$ (the 3rd dimension remains constant because of no gradient)
> >
> > Therefore, we have a constant gap with a distance of 1.0, which is also orthogonal to the embedding spans.
> >
> > **Is closeness of features from different modalities indicate a better generation performance?**
> >
> > > The paper aims to mitigate the modality gap between feature space of different modalities, but in the generation task or encoder/decoder-based architecture, is that the closeness of the features from different modalities indicate a better generation performance? I think it would be better to refer to previous works or conduct this kind of experiment to show the assumption of this paper is true in real applications.
> > >
> >
> > Thank you for your suggestion. We added an experiment to verify this assumption. We train a GPT-2 text generator (image captioner) upon CLIP’s text embeddings $x$. During inference, we manually shift all the $x$ to $x+c$ to simulate the modality gap (a constant vector orthogonal to original spans). We report the captioning performance in terms of gap distance $\|c|\$ in the table below. We observe there are substantial performance drops when the gap grows, showing the need to align embeddings and close the gap. We have now included this discussion in Appendix G.
> >
> > | Gap Distance $\|c\|$ | ROUGE-1 | ROUGE-L | METEOR |
> > | --- | --- | --- | --- |
> > | 0.0 | 85.5 | 81.2 | 83.1 |
> > | 0.2 | 76.3 | 71.5 | 73.8 |
> > | 0.4 | 55.8 | 50.6 | 52.5 |
> > | 0.6 | 40.6 | 35.9 | 36.6 |
> > | 0.8 | 30.5 | 26.7 | 26.5 |
> > | 1.0 | 24.1 | 21.0 | 20.3 |
> > | 1.5 | 16.6 | 14.4 | 13.8 |
> > | 2.0 | 13.8 | 12.1 | 11.6 |
> >
> > We again thank Reviewer XDw2 for their review of our manuscript, which is very helpful in improving the paper. We hope the above responses and changes to our manuscript adequately address your concerns, and that you may be willing to improve your rating as a result. Please let us know if you have further questions or concerns!

---

> > > ### Comment · Reviewer_XDw2 · 2023-11-21
> > > **Reply**
> > >
> > > Thanks for the explanation and demonstration.
> > >
> > > I think some issues have been addressed well like the W1,3, Q2.
> > >
> > > For W2, I think more analysis both in terms of methodology and experiments that can explain why corrupt is so effective will be helpful for me to understand the question.
> > >
> > > For Q1, I realize why the modality gap is constant if the image and text share no effective dimensions. And the example just follows that there are no overlapped effective dimensions. But are that some propositions to guarantee this phenomenon?
> > >
> > > Overall, I will change the score from 5 to 6 based on the current rebuttal.

---

> ### Author Response · Authors · 2023-11-22
>
> Dear reviewer XDw2,
>
> Thank you again for providing us with constructive feedback. We appreciate your consideration of our response and willingness to adjust your rating based on our improvements!
>
> Here we provide additional clarification to address your concerns:
>
> > For Q1, I realize why the modality gap is constant if the image and text share no effective dimensions. And the example just follows that there are no overlapped effective dimensions. But are that some propositions to guarantee this phenomenon?
> >
>
> Thanks for the question. We hope to clarify that **the theory still works when image and text share effective dimensions**.
>
> Here we provide a modified example to help understand the process:
>
> Suppose there are two image-text pairs in 3D space; both the 1st and 2nd dimensions of the image embeddings are effective, and the same applies to text embeddings, and their 3rd dimension both are ineffective.
>
> During initialization, let us denote these embeddings as $x_1, x_2, y_1, y_2 \in \mathbb{R}^3$:
>
> $x_1 = [0.7, 0.5, -0.5]$ (the 3rd dimensions are constant across $x_i$)
>
> $x_2 = [-0.7, -0.5, -0.5]$ (the 3rd dimensions are constant across $x_i$)
>
> $y_1 = [0.5, 0.7, 0.5]$ (the 3rd dimensions are constant across $y_i$)
>
> $y_2 = [-0.5, -0.7, 0.5]$ (the 3rd dimensions are constant across $y_i$)
>
> During optimization, only the union of effective dimensions (i.e., 1st and 2nd) will be aligned, and the intersection of ineffective dimensions (i.e., 3rd) will remain constant:
>
> $x_1 = [0.6, 0.6, -0.5]$ (the 3rd dimension remains constant because of no gradient)
>
> $x_2 = [-0.6, -0.6, -0.5]$ (the 3rd dimension remains constant because of no gradient)
>
> $y_1 = [0.6, 0.6, 0.5]$ (the 3rd dimension remains constant because of no gradient)
>
> $y_2 = [-0.6, -0.6, 0.5]$ (the 3rd dimension remains constant because of no gradient)
>
> Therefore, we have a constant gap with a distance of 1.0, which is also orthogonal to the embedding spans.
>
> Instead, **the key assumption of the theory is that image and text share ineffective dimensions, which is empirically verified in various models such as CLIP.** For example, in CLIP, the image has only 25 effective dimensions, and the text has only 230 effective dimensions. Therefore, **the image and text must share ineffective dimensions in 512D space**, as they sum up to at most 255 effective dimensions. Therefore, these ineffective dimensions will be fully determined by initialization, and will not change during optimization, creating a constant orthogonal modality gap.
>
> Please let us know if you have further questions or concerns!

---

> > ### Author Response · Authors · 2023-11-22
> >
> > Thanks for your suggestion to analyze why corrupt is so effective. We have now added a new experiment to explain this and will include it in the final draft.
> >
> > **Why corrupt $C_2^2$ is so effective?**
> >
> > > I think more analysis both in terms of methodology and experiments that can explain why corrupt is so effective will be helpful for me to understand the question.
> > >
> >
> > As we hypothesize that the greater effectiveness of $C_2^2$ is because $C_2^2$ has two effects: 1) **injecting noise in the span** to mitigate alignment noise; 2) **injecting noise in the modality gap direction** to mitigate the model's sensitivity to this gap, we have added a new experiment to verify this hypothesis.
> >
> > When adding noise sampled from Gaussian distributions, we **remove its component in the modality gap direction**. Specifically, given $\epsilon \sim \mathcal{N}(0, \sigma^2 I)$, we compute its projection on the gap direction as $\epsilon_g = \frac{\epsilon \cdot g}{||g||} \frac{g}{||g||}$, where $\frac{g}{||g||}$ is the modality gap direction, then we remove this projection to get a new noise $\epsilon' = \epsilon - \epsilon_g$. We add this new noise during training and name this experiment as **$C^2_2$ (span noise only)**.
> >
> > From the Table below, we see that adding noise only in the span (i.e., $C^2_2$ (span noise only)) makes its performance much worse than adding noise to all the directions (i.e., $C^2_2$), and its performance is similar to removing the modality gap (i.e., $C^2_1$). **Therefore, adding noise (i.e., *corrupt*) actually leads to a similar improvement to removing the modality gap (i.e., *collapse*).** The reason for the greater effectiveness of *corrupt* than *collapse is* that injecting Gaussian noise not only adds noise in the span but also to the modality gap direction.
> >
> > Given the substantial size of the modality gap, adding noise only is not enough to fully diminish the gap, and adding noise and removing the gap (i.e., $C^3$) still enhances the overall performance.
> >
> > |  | BLEU-1 | BLEU-4 | METEOR | ROUGE-L | CIDEr | SPICE |
> > | --- | --- | --- | --- | --- | --- | --- |
> > | $C^1$ | 28.1 | 2.4 | 12.2 | 25.4 | 13.0 | 6.8 |
> > | $C^2_1$ | 44.4 | 6.1 | 15.5 | 33.6 | 25.2 | 9.2 |
> > | **$C^2_2$ (span noise only)** | 41.2 | 6.2 | 14.9 | 33.6 | 22.8 | 8.3 |
> > | $C^2_2$  | 69.0 | 25.5 | 24.3 | 50.8 | 87.6 | 17.6 |
> > | $C^3$ | 71.0 | 27.7 | 25.0 | 52.0 | 93.3 | 18.3 |
> >
> > We thank again for Reviewer XDw2's great questions. We hope this new experiment fully addresses your concerns.

---

### Official Review · Reviewer_CpsN · 2023-10-29

**Soundness:** 4 excellent
**Presentation:** 4 excellent
**Contribution:** 3 good
**Rating:** 8
**Confidence:** 4

**Summary:**

The paper provides a theoretical understanding of the geometry of the multi-modal contrastive representation space, which is related to the modality gap and alignment noise. Based on this, it presents a new 3-stage framework, called C3 (Connect, Collapse, Corrupt), for solving cross-modal tasks using single-modal data. C3 can effectively bridge the modality gap and enhance the interchangeability of embeddings in the shared representation space. The paper demonstrates the empirical effectiveness of C3 by showing that it achieves state-of-the-art results in various cross-modal tasks.

**Strengths:**

1. Theoretical Insight: The paper's theoretical understanding of the geometry of the multi-modal contrastive representation space is a significant contribution. It also helps shed light on the challenges related to the modality gap (by showing that it is attributed to the joint effect of initialization and optimization), which is the key issue in multi-modal/cross-modal learning.

2. C3 Algorithm: The rationale behind the proposed C3 method is sound and well-explained. Even though each individual step has been explored by previous work, the combination of them leads to very competitive performance on a variety of tasks compared with recent strong baselines (as shown in Table 2-3). The paper also provides comprehensive ablation studies and qualitative examples to understand the effect of each component.

3. Presentation: The presentation is clear, and the ideas are easy to follow. The visuals also help illustrate the effectiveness of the proposed method. The current submission does not include code, hopefully the authors can release them later to facilitate future research.

**Weaknesses:**

The proposed C3 algorithm has limited novelty on its own given that each step has been studied in previous work. However, the combination of these steps is new and well-motivated by the theoretical framework developed in this paper, which mitigates the lack-of-novelty issue.

**Questions:**

1. How is the "Collapse" step implemented (i.e., computing e_x' and e_y')? Is it the same as batchnorm?

---

> ### Author Response · Authors · 2023-11-19
>
> We thank Reviewer CpsN for their positive comments and for providing thoughtful feedback on our work. We address Reviewer CpsN’s concerns below.
>
> **Novelty of $C^3$ algorithm**
>
> > The proposed $C^3$ algorithm has limited novelty on its own given that each step has been studied in previous work. However, the combination of these steps is new and well-motivated by the theoretical framework developed in this paper, which mitigates the lack-of-novelty issue.
>
> Thank you for the question. Yes, as you mentioned, while existing works have empirically proposed collapse and corrupt operation separately, we differ from them in three aspects:
>
> 1. Our method is built on the theoretical understanding of multi-modal contrastive representation space geometry.
> 2. We unified these insights and showed the combination of them leads to superior performance than each of them.
> 3. We proved the effectiveness of our method in a wide range of tasks, modalities, data, and embedding spaces, which can become a standard recipe for all future works built on multi-modal contrastive embedding space.
>
> **Collapse operation implementation**
>
> > How is the "Collapse" step implemented (i.e., computing e_x' and e_y')? Is it the same as batchnorm?
>
> Thanks for the question. The “collapse” step can be viewed as a distribution norm, where the image embedding mean $\bar{e}_x$ and text embedding mean $\bar{e}_y$ are pre-computed on the entire training set and subtracted during training and inference. The batch norm is an approximation with a larger variance. We provided an algorithm formulation of $C^3$ in Appendix C for further clarification.
>
> **Codebase**
>
> > The current submission does not include code, hopefully the authors can release them later to facilitate future research.
>
> Thanks for the question. We included an anonymous GitHub link in the Page 10 reproducibility statement, which should be able to reproduce all the results. Reviewer Vv1o has checked this codebase and commented that “the codebase looks carefully developed and seems free from glaring bugs”.
>
> Thank you again for your feedback, which is very helpful in improving the paper. We hope the above responses and changes to our manuscript adequately address your concerns. Please let us know if you have further questions or concerns!

---

> > ### Comment · Reviewer_CpsN · 2023-11-22
> >
> > I appreciate the authors' response and my concerns are addressed. I'm happy to keep my rating.

---

> > > ### Author Response · Authors · 2023-11-23
> > >
> > > Dear reviewer CpsN,
> > >
> > > We are glad to hear that our response has addressed your concerns. Thank you again for providing insightful reviews and constructive comments!

---

### Official Review · Reviewer_Vv1o · 2023-10-31

**Soundness:** 3 good
**Presentation:** 4 excellent
**Contribution:** 3 good
**Rating:** 8
**Confidence:** 3

**Summary:**

The paper investigates the geometry of embedding spaces obtained through multi-modal contrastive learning (e.g. CLIP), collecting interesting insights and using these to motivate a simple-yet-effective 3-step approach to improve the performance of cross-modal tasks learned using uni-modal data.  In particular, the study suggests, both empirically and theoretically, that the difference between embeddings from different modalities originates from two components: i) a modality gap caused by ineffective dimensions being initialized differently in the two modalities and remaining constant during optimization, and ii) alignment noise that can be approximated as gaussian. The paper then suggests reducing this gap by centering the embeddings at both training time and inference time and adding noise during training. Experiments finally show that the suggested modifications result in state-of-the-art results across a wide set of cross-modal tasks.

**Strengths:**

### Originality

- The paper studies the poorly-understood geometry of latent spaces obtained through multi-modal contrastive learning, building on top of existing works and integrating new insights into an overall recipe to improve cross-modal tasks using such spaces.
- While two out of 3 steps in the CCC method are not novel, the motivation behind the overall framework is a valid contribution, as well as the need of collapsing the ineffective dimensions by subtracting the mean to the embeddings.
- The finding of the modality gap being orthogonal to the text and image spaces is interesting and well motivated.

### Clarity

- The paper is well written and pleasing to read.
- The concepts are explained both rigorously and more colloquially.
- The experiments are well motivated, and their results properly discussed.

### Significance

- The theoretical framework looks solid: the difference of initialization and the lack of gradients for the ineffective dimensions convincingly explains the modality gap.
- The empirical analyses make intuitive sense.
- The framework results in improvements over a wide set of tasks (image/audio/video captioning and text-to-image generation), proving its general applicability.
- The codebase looks carefully developed and seems free from glaring bugs.

Overall, the paper tackles an extremely interesting question that many practitioners share: “how does, and possibly how should, a multi-modal space look like?” and attempt to characterize its geometry with simple yet convincing tools. Both the theoretical and empirical analyses make intuitive sense, and the empirical results on the considered tasks confirm the utility of its findings.

**Weaknesses:**

- The discussion on the alignment noise could be improved: in particular, the results in Table 1 are left for the reader to infer. The same statistics could also be easily computed upon any other modality combination in the appendix, it would be useful to see if it still applies.

**Questions:**

- Since the modality gap is due to the dimensional collapse, would reducing the dimensionality to the effective one help overcoming the issue?
- Is there any relation between the decomposition of the modality gap with the content-style-modality specific decomposition assumed e.g. in [1]? Briefly, each latent vector in a multi-modal contrastive learning space is there assumed to have a part that is shared across modalities, i.e. the content, one that is shared but with some distortion, i.e. the style, and one that is not shared at all, i.e. the modality-specific component. Is it possible that the modality specific component in [1] is just the constant component caused by the different initializations seen in this work?
- The solution to the modality gap is to center the embeddings, implying the modality gap is just a shift. Isn’t it possible that the difference in modality may also result in different scales?

[1] Daunhawer, I., Bizeul, A., Palumbo, E., Marx, A., & Vogt, J. E. (2022, September). Identifiability Results for Multimodal Contrastive Learning. In The Eleventh International Conference on Learning Representations.

---

> ### Author Response · Authors · 2023-11-19
>
> We thank Reviewer Vv1o for their positive comments and for providing thoughtful feedback on our work. We address Reviewer Vv1o’s concerns below.
>
> **More discussion about Table 1**
>
> > The discussion on the alignment noise could be improved: in particular, the results in Table 1 are left for the reader to infer. The same statistics could also be easily computed upon any other modality combination in the appendix, it would be useful to see if it still applies.
>
> Thank you for your suggestion. We have now included a detailed explanation of how to interpret these statistics in Appendix C. We also included statistics of embeddings from different modalities in Appendix C. We hope these edits will help readers easily understand Table 1.
>
>
> **Would reduce the dimensionality mitigate the modality gap?**
>
> > Since the modality gap is due to the dimensional collapse, would reducing the dimensionality to the effective one help overcoming the issue?
>
> Thanks for the question. To understand how dimensionality affects the gap, we initialized CLIP with different dimensions. We find that, in lower dimensions, the ratio of effective dimensions increases for both encoders (but cannot reach 100%) and the modality gap decreases.
>
> | Dimension | Effective Image Dimension | Effective Text Dimension | Init Gap Distance |
> | --- | --- | --- | --- |
> | 64 | 18 | 60 | 0.386 |
> | 128 | 20 | 110 | 0.554 |
> | 256 | 23 | 176 | 0.803 |
> | 512 | 25 | 230 | 1.136 |
>
> Nonetheless, we hope to clarify that while the relationship between the dimensionality and the gap is an interesting question, this gap can be well addressed by the collapse operation based on our theoretical analysis. Therefore, the gap actually does not matter.
>
> **Connection to [1]**
>
> > Is there any relation between the decomposition of the modality gap with the content-style-modality specific decomposition assumed e.g. in [1]? Briefly, each latent vector in a multi-modal contrastive learning space is there assumed to have a part that is shared across modalities, i.e. the content, one that is shared but with some distortion, i.e. the style, and one that is not shared at all, i.e. the modality-specific component. Is it possible that the modality specific component in [1] is just the constant component caused by the different initializations seen in this work?
>
> Thank you for pointing out this interesting work. The modality-specific component in [1] appears to be more closely related to the "alignment noise" than the "modality gap". For instance, [1] identifies "object rotation" as a modality-specific component in the visual domain. This suggests that changes like rotating an image would solely affect image embeddings, with no parallel effect in text embeddings. Therefore, this phenomenon seems to align more with alignment noise, where image and text embeddings are not perfectly aligned in the shared representation space.
>
> **Can modality gap differ in scale?**
>
> > The solution to the modality gap is to center the embeddings, implying the modality gap is just a shift. Isn’t it possible that the difference in modality may also result in different scales?
>
> Thanks for your question. Based on our theoretical analysis, the modality gap can only be a shift, not a scale difference. Our analysis reveals that the embeddings’ effective dimensions from different modalities will be aligned, and the ineffective dimensions will remain constant, which creates a constant gap. If there is a scale difference, gradients will be propagated into the effective dimensions and make the scale the same.
>
> Thank you again for your feedback, which is very helpful in improving the paper. We hope the above responses and changes to our manuscript adequately address your concerns. Please let us know if you have further questions or concerns!
>
> **References**
>
> [1] Daunhawer, I., Bizeul, A., Palumbo, E., Marx, A., & Vogt, J. E. (2022, September). Identifiability Results for Multimodal Contrastive Learning. In The Eleventh International Conference on Learning Representations.

---

> > ### Comment · Reviewer_Vv1o · 2023-11-20
> >
> > I thank the authors for their response. I find my concerns to be addressed, and confirm my Accept score.

---

> > > ### Author Response · Authors · 2023-11-22
> > >
> > > Dear reviewer Vv1o,
> > >
> > > We are glad to hear that our response has addressed your concerns. Thank you again for providing insightful reviews and constructive comments!

---

### Official Review · Reviewer_bNKP · 2023-11-01

**Soundness:** 3 good
**Presentation:** 3 good
**Contribution:** 2 fair
**Rating:** 6
**Confidence:** 3

**Summary:**

This paper explores the modality gap phenomenon in multimodal learning. Specifically, the authors claim that the modality gap emerges and is preserved due to a) dimensional collapse during initialization and training, and b) alignment noise controlled by temperature. To overcome the modality gap, this paper proposes the C^3 paradigm by subtracting mean of features and add Gaussian noise before decoding the features. Experiments on four tasks involving three modalities show the efficacy of C^3.

**Strengths:**

1.	The paper is well written and easy to follow. The empirical analysis well presents and supports the claims on modality gap, a significant problem in multimodal learning.
2.	Experiments are extensive. The authors experiment on image, text and audio modalities, and the results prove the method is applicable across various tasks.

**Weaknesses:**

1.	My major concern is about novelty. [1] has pointed out that random initialization and contrastive learning causes and preserves modality gap, and [2] has modeled modality gap as a constant orthogonal to image and text span. The proposed C^3 is also an ensemble of existing methods [2][3][4]. Especially, the cross-modal transferability in [2] seems quite similar to the ``interchangeability of embeddings from different modalities’’ in this paper. Please justify.
2.	This paper proposes to align representations from different modalities but without convincing justification. In fact, it remains uncertain what effects are relevant to aligning modalities. [1] reports that making the modality gap too small or too large harms performance. [5] proves that strictly aligning modality representations is suboptimal. Therefore, I suggest adding reasons for aligning modalities.
3.	Despite that the authors have conducted experiments on various tasks, the comparison with existing methods is limited. Most comparisons in this paper are ablating over different components in C^3. Tab.2 shows marginal improvement over CapDec without reporting std over independent runs, which is not convincing. Tab.3, Tab.4 and Tab.5 report few comparisons with SOTA methods.

[1] Liang, Victor Weixin, et al. "Mind the gap: Understanding the modality gap in multi-modal contrastive representation learning."
[2] Zhang, Yuhui, et al. "Diagnosing and rectifying vision models using language."
[3] Radford, Alec, et al. "Learning transferable visual models from natural language supervision."
[4] Zhou, Yufan, et al. "Towards language-free training for text-to-image generation."
[5] Jiang, Qian, et al. "Understanding and constructing latent modality structures in multi-modal representation learning."

**Questions:**

1.	From the experiment results, $C_2^2$ seems to be much more effective than $C_1^2$. Why?
2.	Section 3.2 mentions the effect of temperature, but no discussion is given in experiments. What are the effects of modifying temperature in stage 1 and std of Gaussian in stage 3?

---

> ### Author Response · Authors · 2023-11-19
>
> We thank Reviewer bNKP for their positive comments and for providing thoughtful feedback on our work. We address Reviewer bNKP’s concerns below.
>
> **Novelty compared to existing works**
>
> > My major concern is about novelty. [1] has pointed out that random initialization and contrastive learning causes and preserves modality gap, and [2] has modeled modality gap as a constant orthogonal to image and text span. The proposed C^3 is also an ensemble of existing methods [2][3][4]. Especially, the cross-modal transferability in [2] seems quite similar to the ``interchangeability of embeddings from different modalities’’ in this paper. Please justify.
>
> Thank you for the question. While some previous works have found the modality gap exists and empirically observed that addressing the gap can improve performance, no paper so far can provide **a comprehensive and complete theoretical understanding of why there is a gap, how the geometry looks like, and what we should do for the gap**. In this work, we provide a unified theoretical understanding of the gap and provide a simple yet effective method that addresses the gap in a principled way. Our experiments demonstrate its effectiveness across a wide range of tasks, which can be a template for future work.
>
> Here we further provide a detailed comparison with previous works, which is also included in Appendix A:
>
> In terms of **theory** about multi-modal representation space geometry:
>
> [1] explained there is a modality gap caused by initialization and optimization, but its theory failed to justify its geometry.
>
> [2] empirically identified the modality gap geometry, but it failed to explain how the geometry arises.
>
> We are the first work that explains how the geometry arises. We identified the *dimensional collapse* phenomenon in initialization and the *collapsed gradients* in optimization, both of which are not explored by [1,2], and their combination well explained that the modality gap geometry would be a constant orthogonal vector. We also connect *dimensional collapse* back to the *cone effect* proposed in [1] (Appendix F.2), while the former is a more general and well-explored analytical method.
>
> In terms of **methods** for enhancing the interchangeability of embeddings from different modalities:
>
> [2] proposed the *collapse* operation (i.e., removing embedding mean from each modality) in classification settings, based on an empirical understanding of the modality gap geometry.
>
> [3,4] empirically found that the *corrupt* operation (i.e., adding noise to embeddings) works in generation settings.
>
> Our work unified these insights and showed the combination of them leads to superior performance than each of them. Our method is built on our solid theoretical understanding of multi-modal representation space geometry. We proved our method's effectiveness in a wide range of tasks, modalities, data, and embedding spaces, which can become a standard recipe for all future works built on multi-modal contrastive embedding space.
>
> In terms of the **relation** between *cross-modal transferability* and *embedding interchangeability*:
>
> *Embedding interchangeability* is a more fundamental concept that enables *cross-modal transferability* on both classifiers (demonstrated in [2]) and generators (shown in this work). Therefore, in this work we focus on enhancing embedding interchangeability.

---

> > ### Author Response · Authors · 2023-11-19
> >
> > **Why aligning representations from different modalities?**
> >
> > > This paper proposes to align representations from different modalities but without convincing justification. In fact, it remains uncertain what effects are relevant to aligning modalities. [1] reports that making the modality gap too small or too large harms performance. [5] proves that strictly aligning modality representations is suboptimal. Therefore, I suggest adding reasons for aligning modalities.
> >
> > Thank you for the suggestion. If embeddings from different modalities are aligned, we can train a model on one modality and then infer on another modality, enabling us to build cross-modal applications with only uni-modal data. This is an emerging field that lacks principled approaches to be easily applied without requiring more empirical tuning.
> >
> > [1,5] did not find aligning embeddings to be helpful because they use zero-shot retrieval settings where no model further consumes these embeddings. In contrast, [2,3,4] has a classifier, text generator, and image generator trained on these embeddings. Therefore, switching modality embeddings in [2,3,4] without aligning them will suffer from a significant performance drop.
> >
> > We added an experiment to explain this further. We train a text generator (image captioner) over CLIP’s text embeddings $x$. During inference, we manually shift all the $x$ to $x+c$ to simulate the modality gap (a constant vector orthogonal to original spans). We report the captioning performance in terms of gap distance $\|c\|$ in the table below. We observe substantial performance drops when the gap grows, showing the need to align embeddings. We have now included this discussion in Appendix G.
> >
> > | Gap Distance $\|c\|$ | ROUGE-1 | ROUGE-L | METEOR |
> > | --- | --- | --- | --- |
> > | 0.0 | 85.5 | 81.2 | 83.1 |
> > | 0.2 | 76.3 | 71.5 | 73.8 |
> > | 0.4 | 55.8 | 50.6 | 52.5 |
> > | 0.6 | 40.6 | 35.9 | 36.6 |
> > | 0.8 | 30.5 | 26.7 | 26.5 |
> > | 1.0 | 24.1 | 21.0 | 20.3 |
> > | 1.5 | 16.6 | 14.4 | 13.8 |
> > | 2.0 | 13.8 | 12.1 | 11.6 |
> >
> > **Stability of our method and comparison with more baselines**
> >
> > > Despite that the authors have conducted experiments on various tasks, the comparison with existing methods is limited. Most comparisons in this paper are ablating over different components in C^3. Tab.2 shows marginal improvement over CapDec without reporting std over independent runs, which is not convincing. Tab.3, Tab.4 and Tab.5 report few comparisons with SOTA methods.
> >
> > Thank you for the question. We have run our method $C^3$ in Table 2 three times with random seeds 1234, 5678, and 910 and reported the numbers below. We observe a very small variance across different runs, and $C^3$ consistently outperforms all the baselines. We have included these results in Table 2.
> >
> > In terms of baselines, we have included the most comparable baseline Lafite in Table 3. To the best of our knowledge, there is no similar zero-shot audio/video captioning baseline in Table 4.
> >
> > We hope to emphasize that Table 3/4 is mainly used to show the generalization of our method across different tasks, data, modalities, and embedding spaces, where we can see $C^3$ consistently outperforms all the other baselines by a large margin, showing the importance to align embeddings.
> >
> > | Seed | BLEU-1 | BLEU-4 | METEOR | ROUGE-L | CIDER | SPICE |
> > | --- | --- | --- | --- | --- | --- | --- |
> > | 1234 | 71.0 | 27.6 | 25.0 | 52.0 | 93.2 | 18.3 |
> > | 5678 | 71.1 | 27.6 | 24.9 | 52.0 | 93.6 | 18.4 |
> > | 910 | 70.9 | 27.8 | 25.0 | 52.1 | 93.0 | 18.2 |
> > | Avg | 71.0 | 27.7 | 25.0 | 52.0 | 93.3 | 18.3 |
> > | Std | 0.1 | 0.1 | 0.0 | 0.0 | 0.3 | 0.1 |
> >
> > **Why $C_2^2$ (corrupt) seems more effective than $C_1^2$ (collapse)?**
> >
> > > From the experiment results, $C_2^2$ seems to be much more effective than $C_1^2$. Why?
> >
> > Thanks for the question. While $C_2^2$ (corrupt) seems to be more effective than $C_1^2$ (collapse) in Table 2 (image captioning on CLIP embedding space), they seem to be similarly effective in Table 3 (text-to-image generation on CLIP embedding space) and Table 4 (image / video / audio captioning on ImageBind embedding space).
> >
> > We hypothesize that the greater effectiveness of $C_2^2$ is because $C_2^2$ injects random noise in the embedding space during training, including the direction of the modality gap, which potentially reduces the model's sensitivity to this gap. Nevertheless, given the substantial size of the modality gap, $C_1^2$ remains necessary to diminish this gap, thereby enhancing the overall performance of $C^3$.

---

> > > ### Author Response · Authors · 2023-11-19
> > >
> > > **How temperature during multi-modal contrastive learning affects the collapse operation?**
> > >
> > > > Section 3.2 mentions the effect of temperature, but no discussion is given in experiments. What are the effects of modifying temperature in stage 1 and std of Gaussian in stage 3?
> > >
> > > Thanks for the interesting question. We leave this to future work as there are no open-sourced CLIP models trained with different temperatures.
> > >
> > > Thank you again for your feedback, which is very helpful in improving the paper. We hope the above responses and changes to our manuscript adequately address your concerns, and that you may be willing to improve your rating as a result. Please let us know if you have further questions or concerns!
> > >
> > > **References**
> > >
> > > [1] Liang, Victor Weixin, et al. "Mind the gap: Understanding the modality gap in multi-modal contrastive representation learning."
> > >
> > > [2] Zhang, Yuhui, et al. "Diagnosing and rectifying vision models using language."
> > >
> > > [3] Radford, Alec, et al. "Learning transferable visual models from natural language supervision."
> > >
> > > [4] Zhou, Yufan, et al. "Towards language-free training for text-to-image generation."
> > >
> > > [5] Jiang, Qian, et al. "Understanding and constructing latent modality structures in multi-modal representation learning.”

---

> > > > ### Author Response · Authors · 2023-11-22
> > > >
> > > > Dear reviewer bNKP,
> > > >
> > > > We would like to follow up to see if our response addresses your concerns or if you have any further questions. We would really appreciate the opportunity to discuss this further if our response has not already addressed your concerns. Thank you very much!

---

> > > > > ### Author Response · Authors · 2023-11-22
> > > > >
> > > > > Dear reviewer bNKP,
> > > > >
> > > > > We have now added a new experiment to explain **why corrupt $C_2^2$ seems to be more effective** per reviewer XDw2's suggestion. We hope this new experiment provides additional evidence to address your concerns.
> > > > >
> > > > > > From the experiment results, $C_2^2$ seems to be much more effective than $C_1^2$. Why?
> > > > >
> > > > > As we hypothesize that the greater effectiveness of $C_2^2$ is because $C_2^2$ has two effects: 1) **injecting noise in the span** to mitigate alignment noise; 2) **injecting noise in the modality gap direction** to mitigate the model's sensitivity to this gap, we have added a new experiment to verify this hypothesis.
> > > > >
> > > > > When adding noise sampled from Gaussian distributions, we **remove its component in the modality gap direction**. Specifically, given $\epsilon \sim \mathcal{N}(0, \sigma^2 I)$, we compute its projection on the gap direction as $\epsilon_g = \frac{\epsilon \cdot g}{||g||} \frac{g}{||g||}$, where $\frac{g}{||g||}$ is the modality gap direction, then we remove this projection to get a new noise $\epsilon' = \epsilon - \epsilon_g$. We add this new noise during training and name this experiment as **$C^2_2$ (span noise only)**.
> > > > >
> > > > > From the Table below, we see that adding noise only in the span (i.e., $C^2_2$ (span noise only)) makes its performance much worse than adding noise to all the directions (i.e., $C^2_2$), and its performance is similar to removing the modality gap (i.e., $C^2_1$). **Therefore, adding noise (i.e., *corrupt*) actually leads to a similar improvement to removing the modality gap (i.e., *collapse*).** The reason for the greater effectiveness of *corrupt* than *collapse is* that injecting Gaussian noise not only adds noise in the span but also to the modality gap direction.
> > > > >
> > > > > Given the substantial size of the modality gap, adding noise only is not enough to fully diminish the gap, and adding noise and removing the gap (i.e., $C^3$) still enhances the overall performance.
> > > > >
> > > > > |  | BLEU-1 | BLEU-4 | METEOR | ROUGE-L | CIDEr | SPICE |
> > > > > | --- | --- | --- | --- | --- | --- | --- |
> > > > > | $C^1$ | 28.1 | 2.4 | 12.2 | 25.4 | 13.0 | 6.8 |
> > > > > | $C^2_1$ | 44.4 | 6.1 | 15.5 | 33.6 | 25.2 | 9.2 |
> > > > > | **$C^2_2$ (span noise only)** | 41.2 | 6.2 | 14.9 | 33.6 | 22.8 | 8.3 |
> > > > > | $C^2_2$  | 69.0 | 25.5 | 24.3 | 50.8 | 87.6 | 17.6 |
> > > > > | $C^3$ | 71.0 | 27.7 | 25.0 | 52.0 | 93.3 | 18.3 |

---

### Author Response · Authors · 2023-11-19

We thank all the reviewers for their thoughtful and constructive feedback on our manuscript. We are encouraged to hear that reviewers find the problem we studied about multi-modal contrastive representation geometry important and significant (bNKP, Vv1o, CpsN), the theory we proposed to explain the representation geometry novel, sound, and valuable (bNKP, Vv1o, CpsN, XDw2), the experiments and analyses we conducted solid, insightful and well-justified (bNKP, Vv1o, CpsN, XDw2), and the paper well-written and easy-to-follow (bNKP, Vv1o, CpsN, XDw2).

In response to feedback, we provide individual responses below to each reviewer, and **we carefully updated the paper based on the reviewers’ suggestions (updates highlighted in blue)**. We would again like to thank all the reviewers for their time and feedback, and we hope that our responses and the revised manuscript adequately address all the concerns. Please let us know if you have further questions or concerns!

---

### Meta-Review · Area_Chair_oaRK · 2023-12-10

**Metareview:**

This paper investigates the modality gap phenomenon in multimodal learning, assuming it arises due to 1) dimensional collapse during initialization and training, and 2) alignment noise regulated by temperature. To address this gap, the paper proposes the 'Connect, Collapse, Corrupt' paradigm, involving subtracting the mean of features and introducing Gaussian noise before decoding the features.

Following the rebuttal phase, all reviewers unanimously agreed on acceptance as the authors effectively addressed most concerns raised by reviewers. The paper is well-written, providing a theoretical analysis of the modality gap, and leveraging insights from this analysis, it proposes effective methods to alleviate the gap.

One reviewer expressed concerns about the similarity between prior work and this study, suggesting a potential limitation in novelty and contribution. The authors clarified the connections between previous works and their own during the rebuttal phase. We strongly recommend the authors incorporate these discussions into the final version of the paper.

Overall, we recommend accepting this submission for ICLR. Congratulations!

**Justification For Why Not Higher Score:**

Although this work has provided a unified theoretical understanding of the modality gap and a simple yet effective method to address the gap in a principled way, there are already existing works that discovered the modality gap exists and empirically observed that addressing the gap can improve performance.

**Justification For Why Not Lower Score:**

All reviewers agree to accept this work after rebuttal based on its contribution to the community.

---

### Decision · Program_Chairs · 2024-01-16

Accept (poster)